# Long non-coding RNA KIKAT/LINC01061 as a novel epigenetic regulator that relocates KDM4A on chromatin and modulates viral reactivation

Wan-Shan Yang[1,2‡], Wayne W. Yeh[1,2], Mel Campbell[3], Lung Chang👁[4,5], Pei-Ching Chang👁[1,2,6]*

**1** Institute of Microbiology and Immunology, National Yang Ming Chiao Tung University, Hsinchu, Taiwan, **2** Institute of Microbiology and Immunology, National Yang-Ming University, Taipei, Taiwan, **3** UC Davis Cancer Center, University of California, Davis, California, United States of America, **4** Department of Pediatrics, MacKay Children's Hospital and MacKay Memorial Hospital, Taipei, Taiwan, **5** Department of Medicine, MacKay Medical College, New Taipei, Taiwan, **6** Cancer Progression Research Center, National Yang-Ming University, Taipei, Taiwan

‡ First author.
* pcchang@ym.edu.tw

**Data Availability Statement:** All relevant data are within the manuscript and its Supporting Information files.

## Abstract

KDM4A is a histone lysine demethylase that has been described as an oncogene in various types of cancer. The importance of KDM4A-mediated epigenetic regulation in tumorigenesis is just emerging. Here, by using Kaposi's sarcoma associated herpesvirus (KSHV) as a screening model, we identified 6 oncogenic virus-induced long non-coding RNAs (lncRNAs) with the potential to open chromatin. RNA immunoprecipitation revealed KSHV-induced KDM4A-associated transcript (KIKAT)/LINC01061 as a binding partner of KDM4A. Integrated ChIP-seq and RNA-seq analysis showed that the KIKAT/LINC01061 interaction may mediate relocalization of KDM4A from the transcription start site (TSS) of the AMOT promoter region and transactivation of AMOT, an angiostatin binding protein that regulates endothelial cell migration. Knockdown of AMOT diminished the migration ability of uninfected SLK and iSLK-BAC16 cells in response to KIKAT/LINC01061 overexpression. Thus, we conclude that KIKAT/LINC01061 triggered shifting of KDM4A as a potential epigenetic mechanism regulating gene transactivation. Dysregulation of KIKAT/LINC01061 expression may represent a novel pathological mechanism contributing to KDM4A oncogenicity.

## Author summary

Epigenetic regulation of chromatin structure and gene function connects genotype to phenotype and diseases. Long non-coding RNA (lncRNA) is emerging as a novel type of epigenetic regulator exhibiting diverse biological functions. Aberrant lncRNA expression is associated with various diseases, including cancer. The widespread epigenetic changes that occur during the latent-to-lytic switch of herpes virus life cycle make it an attractive

**Funding:** This work was supported by the Ministry of Science and Technology (URL: https://www.most.gov.tw/?l=ch; Grant Number: 108-2320-B-010-029-MY3; 109-2926-I010 503; 109-2811-B-010-543). P-CC received the grants. The funders had no role in study design, data collection and analysis, decision to publish, or preparation of the manuscript.

model to study epigenetic regulation. Using Kaposi's sarcoma associated herpesvirus (KSHV) as a model, we screened the epigenetic function of lncRNAs whose expression was induced by reactivation of this oncogenic virus. KIKAT/LINC01061 was identified as a novel histone lysine-specific demethylase 4A (KDM4A) interacting lncRNA. KDM4A is the first identified histone trimethyl demethylase that has been demonstrated as an oncogene in various cancers. Our data reveal a novel lncRNA-mediated regulation of the epigenetic function of KDM4A and demonstrate this lncRNA-chromatin modifier interaction may serve as a potential target in cancer therapy.

## Introduction

Chromatin compactness, a major determinant of gene activity, is tightly regulated by histone modifications. By recruiting different protein complexes to chromatin, histone methylation can either activate or repress gene expression. Histone lysine (K) residues exist in a mono- (me1), di- (me2) or tri- (me3) methylated state. Since the initial discovery in 2004 of the first histone lysine demethylase (KDM) KDM1A (also known as lysine-specific demethylase 1, LSD1), that removes me2 and me1 from histones, the importance of histone lysine methylation in transcription regulation has received considerable attention [1]. In 2006, the identification of KDM4A (also known as jumonji C (JmjC)-domain-containing histone demethylase 2A, JMJD2A), uncovered a class of α-ketoglutarate-dependent KDMs that are able to remove me3 from histone substrates [2] making histone lysine methylation completely reversible, resulting in even more attention towards the role of histone methylation in transcription regulation.

The impact of KDM4A on epigenetic reprograming has been long investigated focusing on H3K9me3, a histone mark strongly associated with transcriptionally silent heterochromatin. In general, removal of H3K9me3 on a promoter region by KDM4A is associated with gene up-regulation [3–6]. On the other hand, removal of acetylation on the promoter region by KDM4A when complexed with histone deacetylases (HDACs) is associated with gene repression [7–9]. These data together indicate that KDM4A may either activate or repress transcription in different epigenetic contexts. More importantly, one recent report demonstrated that KDM4A not only demethylates H3K9me3 at broad domains of H3K4me3 (bdH3K4me3), but also protects bdH3K4me3 from the invasion of H3K9me3. This protection is critical for gene transactivation during embryo development [10]. However, little is known about KDM4A relocation from promoter region to barrier sites.

By using Kaposi's sarcoma associated herpesvirus (KSHV) as a model to study epigenetic regulation, we have previously shown that KDM4A is essential for gene activation [11] and post-translational modification of KDM4A by small ubiquitin-like modifier (SUMO) is important for KDM4A binding on chromatin and gene transactivation [12]. In spite of cumulative findings, the underlying mechanism for KDM4A-mediated epigenetic regulation of transcription remains unclear. Recently, long non-coding RNA (lncRNA), defined as RNA transcripts > 200 nucleotides (nt) that do not possess protein-coding capabilities, is emerging as a new epigenetic component. The epigenetic regulatory mechanisms ascribed to lncRNAs have been proposed as binding to DNA/chromatin binding proteins and modification enzymes to serve as guides, scaffolds or decoys for these complexes [13]. For example, HOX transcript antisense intergenic RNA (HOTAIR), one of the first lncRNAs described in tumor progression, was identified as a scaffold for histone modification complexes of the histone methyltransferase polycomb repressor complex 2 (PRC2) and the histone demethylase

LSD1-CoREST-complex [14]. Whether and how lncRNA is involved in KDM4A-mediated epigenetic regulation of transcription has never been studied.

Encouraged by our previous reports using KSHV as an experimental model to reveal the epigenetic regulatory function of KDM4A [11] and SUMO modification [12], here, we use the same strategy to identify novel KDM4A-associated epigenetic regulatory lncRNAs. Using combined transcriptome analysis and small interfering RNA (siRNA) knockdown screening approaches, we surveyed 82 KSHV reactivation-induced lncRNAs and identified 6 with the potential of opening the chromatin. By using an RNA immunoprecipitation (RIP) assay, we found KIKAT/LINC01061 as a novel KDM4A-interacting lncRNA. Further bioinformatic analyses revealed KIKAT/LINC01061 as a potential onco-lncRNA. Notably, our study showed that KIKAT/LINC01061 promoted cell migration by up-regulation of AMOT. This process was mediated by relocation of KDM4A within the AMOT promoter, leading to the activation of gene expression. This novel finding extends the epigenetic regulatory network of lncRNAs and histone demethylases in transcription regulation and oncogenesis.

## Results

### Identification of epigenetic regulatory lncRNAs using KSHV reactivation as a model

KSHV has distinct latent and lytic phases whose interconversion is primarily regulated by epigenetic mechanisms. We have previously used KSHV as a model to elucidate the epigenetic function of the histone lysine demethylase KDM4A in KSHV reactivation and cell proliferation [11]. LncRNA is a new epigenetic component that merits further investigation. Therefore, we again used KSHV as a model to identify lncRNAs with epigenetic functions. To identify KSHV reactivation induced lncRNAs, high-throughput RNA sequencing (RNA-seq) was performed using iSLK-BAC16 cell line, in which K-Rta is ectopically expressed in a doxycycline (Dox)-inducible manner. Total RNA prepared from iSLK-BAC16 cells treated with Dox for 24 hours, mimicking K-Rta-induced KSHV reactivation, was subjected to paired-end RNA-seq. Following aligned the raw reads to human genome version GRCh37 using CLC Genomics Workbench 11 and annotated transcripts by RefSeq using Partek Genomic Suite (PGS) v.7.0, we identified 13,241 mRNAs and 1,721 lncRNAs (reads per kilobase of transcript per million mapped reads (RPKM) $\geq$ 0.05) that were expressed in latent or lytic iSLK-BAC16 cells. When the expression profiles between KSHV latent and lytic reactivated iSLK-BAC16 cells were compared, 4,198 (31.7%) mRNAs and 1,025 (59.6%) lncRNAs were identified as differentially expressed by viral reactivation. Among the 505 up-regulated lncRNAs, 59 (12%) were sense, 175 (35%) were antisense, and 215 (43%) were intergenic (long intergenic non-coding RNAs, lincRNAs) (Fig 1A). In this study, we focused on lincRNAs, a subgroup of lncRNAs that do not share sequence with coding loci. To assess the lncRNAs governing the latent-to-lytic conversion, we performed siRNA screening using the commercial Human Lincode siRNA Library-NR lncRNA RefSeq v54-SMARTpool (Dharmacon) that contains 4 siRNAs targeting different regions of each lncRNA. Of the 215 KSHV reactivation up-regulated lincRNAs, 82 were present in the siRNA library and screened (Fig 1B).

To monitor the KSHV latent-to-lytic switch, we used an image-based assay relying on observations of the green and red fluorescence signals of rKSHV.219 [15], a dual fluorescence recombinant KSHV (rKSHV) that expresses green fluorescent protein (GFP) during latency and red fluorescent protein (RFP) in the lytic phase. For the screening, rKSHV.219 produced from Vero-rKSHV.219 cells treated with 1.75 mM sodium butyrate (NaB) were collected and used to infect iSLK cells. After infection, iSLK-rKSHV.219 cells were re-seeded, transfected with the siRNAs and then treated with Dox (Fig 1C). By evaluation of the RFP:GFP ratio, we detected 20 lincRNAs whose knockdown significantly reduced RFP positive cells, an indicator

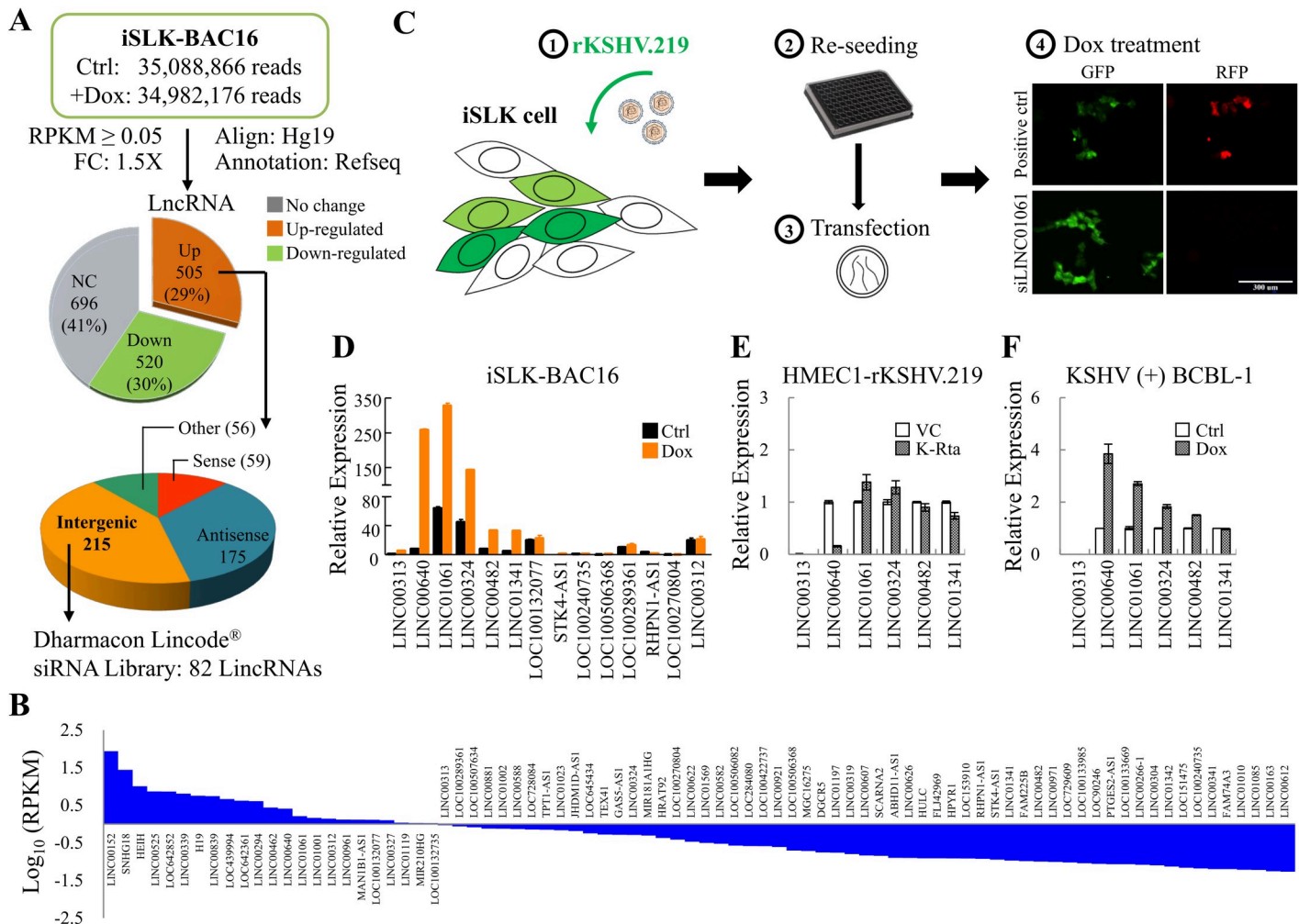

**Fig 1. Transcriptome analysis and siRNA knockdown screening of KSHV reactivation-induced lncRNAs essential for KSHV latent-to-lytic switch. (A)** Workflow showing the identification of KSHV reactivation-induced long intergenic non-coding RNAs (LincRNAs). RNA-seq was performed using total RNA isolated from iSLK-BAC16 cells treated with and without Dox for 24 hours. Paired-end raw reads were aligned to human reference genome hg19 using CLC Genomics Workbench (Qiagen Germantown, MD, USA) and annotated to Refseq using Partek Genomics Suite (Partek, MO, USA). Reads per kilobase of transcript per million mapped read (RPKM) higher than 0.05 in at least one condition was considered as expressed and used for analysis. Pie chart (Top) showing the numbers (percentages) of long non-coding RNAs (LncRNAs) that were up- or down-regulated more than 1.5-fold after Dox induction for viral reactivation. Pie chart (Bottom) showing the numbers of different types of up-regulated lncRNAs. **(B)** Histogram demonstrating the expression (RPKM, log10 scale) of 82 KSHV reactivation up-regulated lincRNAs present in the siRNA Library (Dharmacon, GU-301000). **(C)** Schematic representing the siRNA screening procedure of lincRNAs that mediate KSHV reactivation. The recombinant KSHV, rKSHV.219, produced from Vero-rKSHV.219 cells were used to infect iSLK cells. After 24 hours, iSLK-rKSHV.219 cells were re-seeded in 96-well-plates at a density of 1000 cells/well. Cells were attached overnight and transfected with siRNAs. Eighteen hours after transfection, cells were treated with 2 μg/ml Dox for another 72 hours, fixed by 4% paraformaldehyde, stained with Hochest 33258. The GFP and RFP positive cells were observed with an inverted fluorescence microscope. Representative images (100x magnification) of iSLK-rKSHV.219 cells transfected with siLINC01061 was shown. Cells without siRNA transfection was used as positive control. Scale bar: 300 μm. **(D)** RT-qPCR analysis of the expression of 14 lincRNAs identified in Fig 1C in iSLK-BAC16 cells treated with 1 μg/ml Dox for 24 hours. The expression of lincRNAs was normalized to GAPDH and the relative expression of LINC00313 in iSLK-BAC16 cells (Ctrl) was considered to be 1. **(E)** rKSHV.219 was used to infect HMEC1 cells, followed by transient puromycin selection for a week. After selection, HMEC1-rKSHV.219 cells were re-seeded in 6-well-plates. Cells were attached overnight and transduced with lentiviruses harboring pLenti4-Flag-K-Rta or empty vector control (VC). 72 hours after transduction, total RNA was collected, followed by RT-qPCR analysis of the expression of 6 lincRNAs that were up-regulated in iSLK-BAC16 after viral reactivation. The expression of lincRNAs was normalized to GAPDH and reported as the relative expression of vector control (VC = 1). **(F)** RT-qPCR analysis of the expression of 6 lincRNAs identified in Fig 1D in TREx-F3H3-K-Rta BCBL-1 cells treated with 0.2 μg/ml Dox for 72 hours. The expression of lincRNAs was normalized to GAPDH and plotted as the relative expression versus non-induced cell Ctrl (Ctrl = 1).

of KSHV reactivation, compared with control in three independent screens (S1 Table). Real-time quantitative RT-PCR (RT-qPCR) was only able to detect 14 of the 20 lncRNAs and confirmed a consistent up-regulation of 6 lncRNAs after KSHV reactivation in three independent

experiments (Fig 1D). The expression of the 6 lncRNAs was also examined in rKSHV.219-infected endothelial HMEC1 cells reactivated via K-Rta transduction (Fig 1E) and in the KSHV (+) BCBL-1 cell line, TREx-F3H3-K-Rta BCBL-1 (Fig 1F), following dox-mediated K-Rta induction of KSHV reactivation. Data more consistent with the initial screen were obtained in KSHV-positive BCBL-1 cells than in HMEC1-rKSHV.219 cells, suggesting less effective induction of lncRNAs in transient transduction experiments.

## KIKAT is a novel KDM4A interacting lncRNA

It is our longstanding pursuit to understand the regulatory mechanism of KDM4A in epigenetic regulation of KSHV reactivation. In continuing our research interest, we first chose KDM4A for testing its interaction with the 6 KSHV reactivation-related lncRNAs. To this end, we conducted RIP assays using cell extracts prepared from SLK cells, as these cells expressed higher levels of KDM4A (Fig 2A) than HMEC1 and HUVEC endothelial cells. The KDM4A proteins were immunoprecipitated (IP'd) with anti-KDM4A antibody, and the immunoprecipitates were first probed with anti-KDM4A antibody to confirm the successful pull-down of KDM4A (Fig 2B). To determine if any of the 6 viral reactivation-induced lncRNAs can interact with KDM4A, the RNAs extracted from immunoprecipitates were subjected to RT-qPCR analysis. Notably, an interaction was detectable between KDM4A and LINC01061 (Fig 2C), a lncRNA located on chromosome 4q26 (120,326,678–120,331,815, complement) between PDE5A and FABP2 genes (Fig 2D) and is transcribed as an RNA of approximately 4.8 kb. This interaction was further confirmed in a KSHV containing B lymphoma cell line, TREx-F3H3-K-Rta BCBL-1 cells (Fig 2E and 2F). We therefore conclude that we identified LINC01061 as a KSHV-induced KDM4A-associated transcript (KIKAT).

## Transcriptome analysis identified KIKAT/LINC01061 as a potential onco-lncRNA

To elucidate the cellular function of KIKAT/LINC01061, both transient and stable KIKAT/LINC01061 overexpressed SLK cells were comprehensively searched for potential downstream targets by high throughput RNA-seq. Successful overexpression of KIKAT/LINC01061 in transient transduced SLK cells (Fig 3A) and in a stable SLK-KIKAT/LINC01061 overexpression cell line (Fig 3B) was first checked by RT-qPCR. Following alignment and annotation, we identified 13,298 mRNAs (RPKM ≥ 0.05) that were expressed in SLK cells with or without KIKAT/LINC01061 overexpression (Fig 3C). When the gene expression profiles were compared, approximately 1000 genes were regulated (≥ 2-fold change) by transient KIKAT/LINC01061 overexpression (9%) (S2 Table) and in the SLK-KIKAT/LINC01061 cell line (8.2%) (S3 Table) when compared with controls (Fig 3C).

To elucidate the signaling pathways that are activated by KIKAT/LINC01061, gene ontology (GO) analysis was performed using Ingenuity Pathway Analysis (IPA) software. The genes that are up- and down-regulated by transient and stable overexpression of KIKAT/LINC01061 were both significantly enriched (-log [P-value] >1.3) in cancer-related pathways (Table 1). Among the top 10 up-regulated genes in both transient and stable KIKAT/LINC01061 overexpressed SLK cells (Fig 3D), 7 were in cancer-related pathways, and an up-regulation of 5 of these genes were confirmed in SLK-KIKAT/LINC01061 cell line upon KIKAT/LINC01061 overexpression using RT-qPCR (Fig 3E). Consistent with our hypothesis, the identification of cancer-related pathways in KIKAT/LINC01061 overexpressing cells indicates that an oncogenic virus can be used as a model to identify potential cellular onco-lncRNAs.

To further investigate whether up-regulation of KIKAT/LINC01061 occurred in cancer, the expression levels of KIKAT/LINC01061 was compared in various tumors using a human

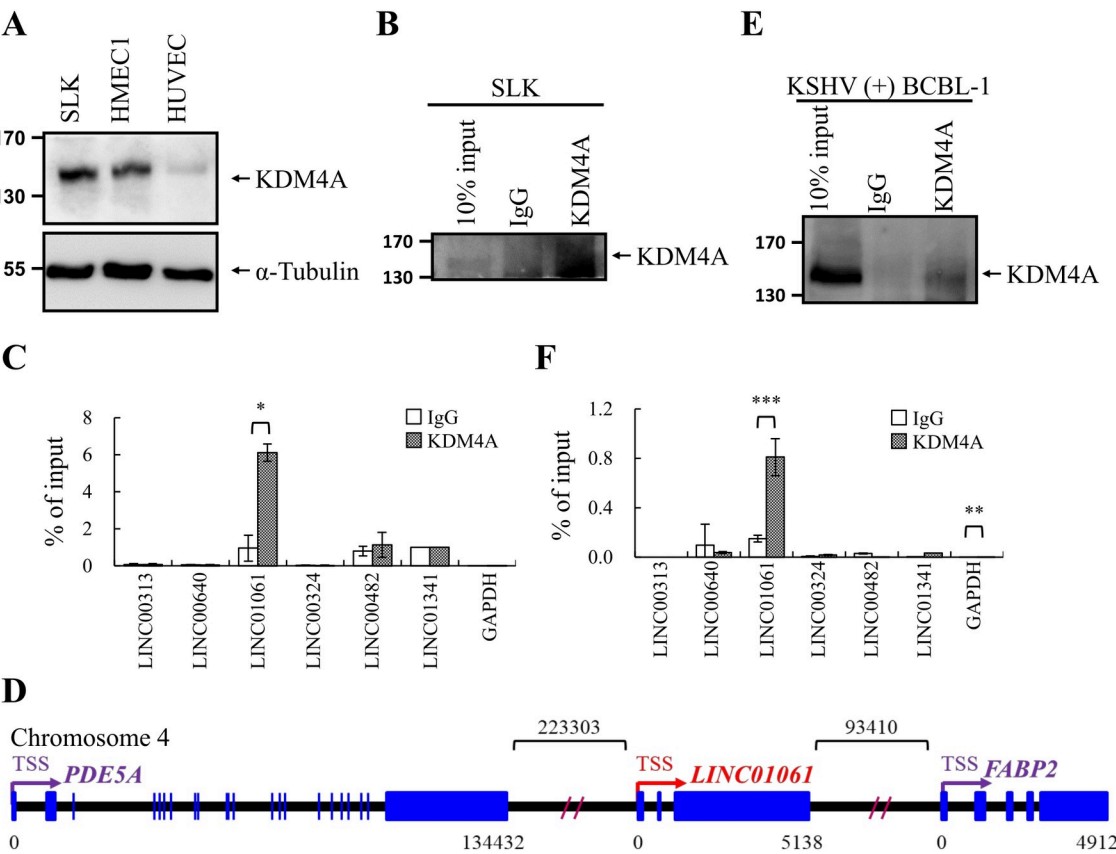

**Fig 2. KSHV-Induced KDM4A-Associated Transcript (KIKAT)/LINC01061 is associated with histone lysine demethylase 4A (KDM4A). (A)** Total cell lysates (TCLs) prepared from SLK, HMEC1 and HUVEC were examined for KDM4A by immunoblotting. Anti-α-Tubulin antibody was included as loading control. **(B)** SLK cell lysate was used for RNA immunoprecipitation (RIP) using KDM4A antibody or rabbit IgG as control. Immunoprecipitation (IP) specificity was confirmed by immunoblotting. **(C)** Copurified RNA from KDM4A and IgG IPs in (B) was assessed for LncRNAs enrichment by RT-qPCR. **(D)** Schematic diagram represented the structure of KIKAT/LINC01061 and the nearby genes Phosphodiesterase 5A (PDE5A) and Fatty acid-binding protein 2 (FABP2). Exons and introns are showed as blue boxes and thin lines, respectively. Arrows labeled transcription start site (TSS) and transcription direction. The numbers below the boxes and above the lines indicate the length of transcripts and of intergenic regions, respectively. **(E)** Cell lysate from TREx-F3H3-K-Rta BCBL-1 cells was used for RIP using KDM4A antibody or rabbit IgG as control. IP specificity was confirmed by immunoblotting. **(F)** Copurified RNA from KDM4A and IgG IPs in (E) was assessed for LncRNAs enrichment by RT-qPCR. Significance was determined by student's $t$-test. $^*p < 0.05$, $^{**}p < 0.01$, $^{***}p < 0.001$.

cancer metastasis database (HCMDB). KIKAT/LINC01061 showed significantly higher expression levels in skin cancer when compared with normal skin (Fig 3F). Moreover, gene set enrichment analysis (GSEA) was also used to assess the association of KIKAT/LINC01061-repressed genes with tumorigenesis. The result showed that the KIKAT/LINC01061 repressed genes were enriched in normal skin with the false discovery rate $q < 0.25$ and $p < 0.05$ (Fig 3G). The correlation between KIKAT/LINC01061 overexpression and Kaposi's sarcoma (KS) lesions was further conducted using GSE147704 and the result showed that KIKAT/LINC01061-repressed genes were also negatively correlated with KS lesions with the false discovery rate $q < 0.25$, while the $p > 0.05$ (Fig 3H).

## Integrated genome-wide chromatin occupancy and expression analyses identify KIKAT/LINC01061 as a novel epigenetic regulator for KDM4A

As KIKAT/LINC01061 is a KDM4A-associated lncRNA, we wanted to assess any changes in chromatin binding of KDM4A in the presence of KIKAT/LINC01061. To achieve this, we

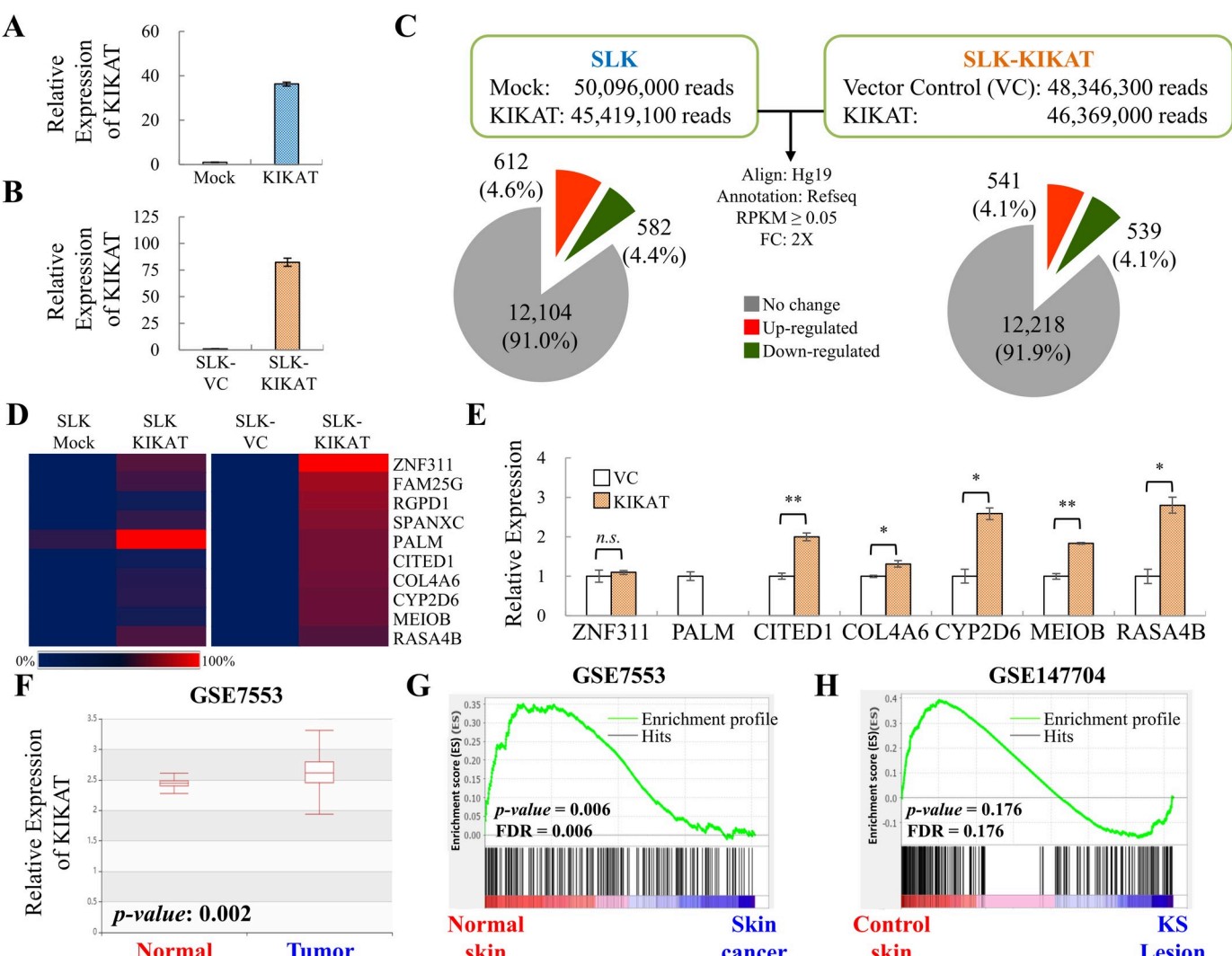

**Fig 3. RNA-Seq analysis revealed that KIKAT/LINC01061 may function as an onco-lncRNA.** (A) RT-qPCR analysis of KIKAT/LINC01061 expression in SLK cells transiently transduced with lentivirus expressing KIKAT/LINC01061 or with no cDNA (mock). (B) RT-qPCR analysis of KIKAT/LINC01061 expression in a stable SLK cell line overexpressing KIKAT/LINC01061, SLK-KIKAT/LINC01061, and its vector control cells, SLK-VC. (C) RNA-seq was performed using total RNA isolated from cells in (A) and (B). Paired-end raw reads were aligned and annotated as described in Fig 1A. RPKM ≥ 0.05 in at least one condition was considered as expressed and used for further analysis. Pie chart showing the numbers (percentages) of mRNAs that were up- or down-regulated more than 2-fold in transient transduced cells (Left) and stable cell lines (Right). (D) Expression heatmap of top 10 up-regulated genes of RNA-seq data from both transient (left) and stable (right) KIKAT/LINC01061 expression SLK cells. (E) RT-qPCR analysis of the expression of top 10 up-regulated genes identified in (D) in SLK-KIKAT/LINC01061 and SLK-VC cells. (F) KIKAT/LINC01061 levels in skin cancer and normal tissues were obtained from the human cancer metastasis database (HCMDB). The statistical significance was calculated using student's *t*-test. (G and H) GSEA analysis reveal the correlation of KIKAT/LINC01061 down-regulated genes (≥ 5-fold) with skin cancer (GSE7553) (G) and Kaposi's sarcoma (KS) lesion (GSE147704) (H). Significance was determined by student's *t*-test. *$p < 0.05$, **$p < 0.01$, *n.s.* non-significant.

generated genome-wide KDM4A binding profiles from SLK-Vector (VC) and SLK-KIKAT/ LINC01061 cells using chromatin immunoprecipitation (ChIP) assays in combination with high-throughput sequencing (ChIP-seq). To obtain a good control for ChIP-seq, we generated a KDM4A knockout SLK cell line, SLK-KDM4A KO, using CRISPR/Cas9n system (S1 Fig) and KDM4A ChIP-seq result from SLK-KDM4A KO was used as reference for peak calling. We obtained approximately 90 million unique aligned reads and identified 21,229 and 20,320 KDM4A-bound regions (FDR ≤ 0.05) in SLK-VC and SLK-KIKAT/LINC01061 cells, respectively (Fig 4A). Peak distribution showed that most of the KDM4A-bound regions were within

**Table 1. Top 10 significant GO categories of KIKAT-regulated genes.**

| Categories/Diseases or Functions Annotation | p-value | |
|---|---|---|
| | Transient | Cell line |
| Cancer, Organismal Injury and Abnormalities | | |
| Non-hematological solid tumor | 1.42E-08 | 5.86E-10 |
| Nonhematologic malignant neoplasm | 2.84E-08 | 3.92E-09 |
| Carcinoma | 1.59E-07 | 8.32E-08 |
| Epithelial neoplasm | 1.60E-07 | 4.58E-08 |
| Tumorigenesis of tissue | 2.05E-07 | 8.60E-08 |
| Non-melanoma solid tumor | 1.41E-06 | 2.48E-08 |
| Malignant solid tumor | 1.34E-05 | 3.81E-07 |
| Cancer | 1.51E-05 | 3.66E-07 |
| Cell-To-Cell Signaling and Interaction | | |
| Communication of cells | 5.39E-06 | - |
| Signal transduction | 5.92E-06 | - |
| Cellular Assembly and Organization, DNA Replication, Recombination, and Repair | | |
| Formation of nucleosomes | - | 3.88E-09 |
| Cancer, Organismal Injury and Abnormalities | | |
| Solid tumor | - | 5.13E-07 |

2 kb of transcription start sites (TSSs) (Fig 4B). Surprisingly, the global distribution of KDM4A-binding regions was largely unaltered upon overexpression of KIKAT/LINC01061 (Fig 4B).

To identify potential target genes for KDM4A, the KDM4A-bound regions were annotated to the promoter region (TSS ± 2,000 bp). The 21,229 and 20,320 KDM4A-bound regions (Fig 4A) in SLK-VC and SLK-KIKAT/LINC01061 cells were annotated to promoter regions of 7,343 and 7,128 genes (Fig 4A), respectively. Consistent with peak distribution (Fig 4B), a large proportion of the potential KDM4A target genes (6,136) were overlapped between control (83.6%) and KIKAT/LINC01061 overexpression (86.0%) (Fig 4A). In combination with gene expression profiles (Fig 4C), most of the KDM4A-binding genes identified by ChIP-seq in both SLK-VC and SLK-KAKIT cells were expressed (4,765 of 6,136; 77.7%) (Fig 4C). However, only approximately 5% (288 of 4,765 genes) of the expressed genes showed either 2-fold up- (145) or down- (143) regulation (Fig 4C and S4 Table) upon KIKAT/LINC01061 overexpression. Together, these data suggest that KIKAT/LINC01061 elicits gene-specific regulatory effects on KDM4A.

## KIKAT/LINC01061-guided KDM4A shifting on chromatin in transcription regulation

KDM4A was initially found to remove repressive histone mark H3K9me3 at promoter regions marked by the active histone mark H3K4me3 [16]. More importantly, Sankar *et al* recently showed that during early development, KDM4A may form reprogramming barriers to protect H3K4me3 from H3K9me3 invasion [10]. However, how KDM4A shifts from TSSs to enable transcription activation is largely unknown. Since lncRNAs can act as guides for chromatin modifying enzymes [13], we hypothesized that the KIKAT/LINC01061 interaction may contribute to the movement of KDM4A at promoter regions during gene activation. Following this hypothesis, we checked KDM4A-binding peaks in promoter regions of the 145 KIKAT/LINC01061 overexpression up-regulated genes and observed a shift (> 300 bp) of KDM4A-binding peaks in 44 of them (Table 2).

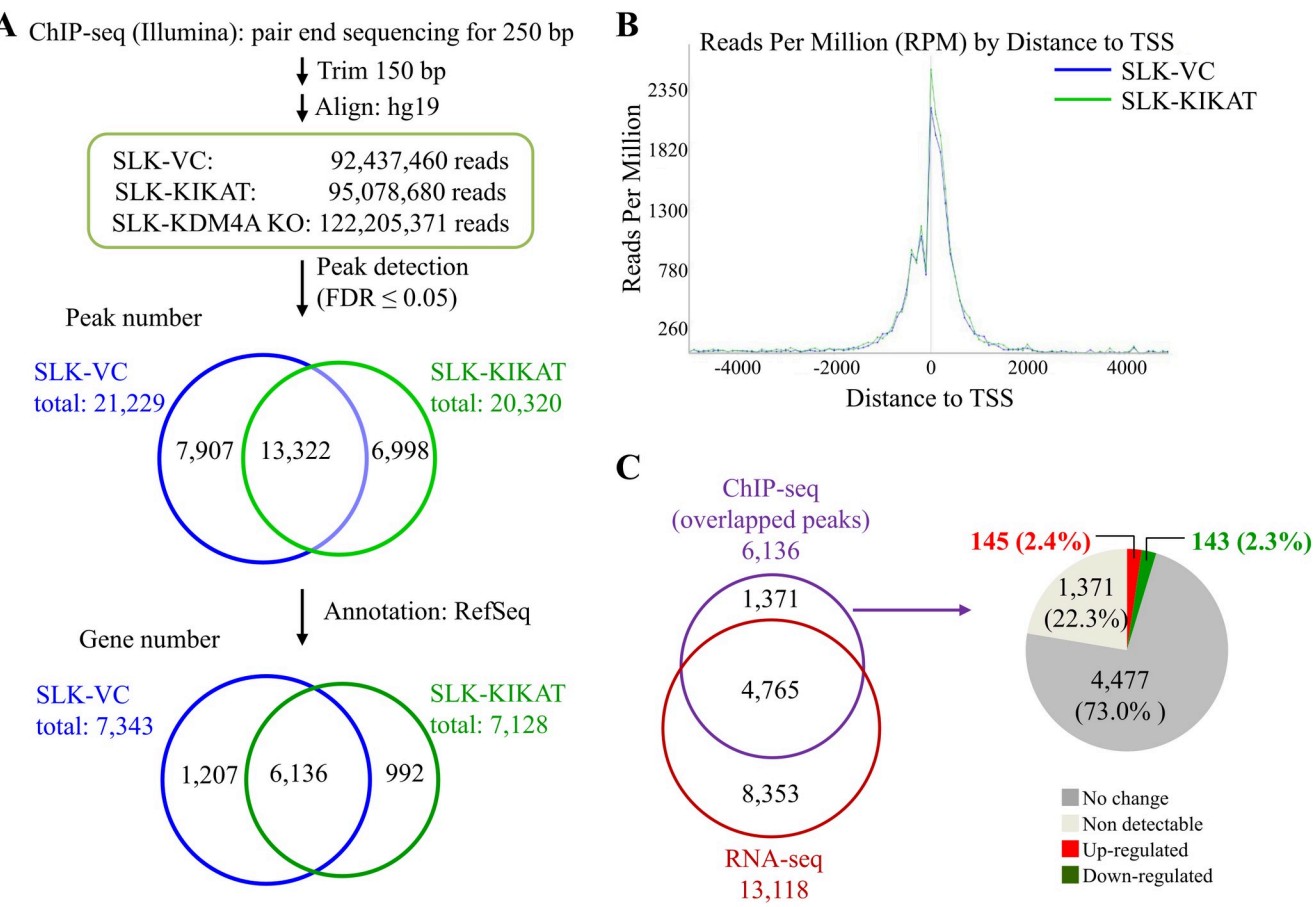

**Fig 4. Genome-wide ChIP-seq analysis revealed the regulatory role of KIKAT/LINC01061 in KDM4A occupancy in SLK cells. (A)** Workflow showing the identification of KDM4A ChIP-seq peaks. ChIP-seq was performed using chromatin DNA immunoprecipitated (IPed) from SLK-VC, SLK-KIKAT/LINC01061 and SLK-KDM4A KO-VC cells by KDM4A antibody. Paired-end raw reads were trimmed and aligned to human reference genome hg19 using CLC Genomics Workbench (Qiagen Germantown, MD, USA). KDM4A ChIP-seq peaks were called using Partek Genomics Suite (Partek, MO, USA) with KDM4A KO ChIP DNA as control. Peaks with FDR ≤ 0.05 were identified as significant KDM4A binding regions. KDM4A peaks were annotated to promoter regions (TSS ± 2 kb) based on RefSeq. Venn diagrams represent the KDM4A peaks (Middle) and potential target genes (bottom) coincide in SLK-VC and SLK-KIKAT/LINC01061. **(B)** Distance distribution of KDM4A peaks. **(C)** Venn diagram depicts the overlap of genes among RNA-seq (Fig 3C, right panel) and ChIP-seq (Fig 4A, bottom panel). Pie chart showing the numbers (percentages) of mRNAs that were up- or down-regulated more than 2-fold in SLK-KIKAT/LINC01061 when compared with SLK-VC.

Of the 44 genes associated with a shift of KDM4A-binding peaks, 14 had been identified in cancer-related pathways (Table 2). Among these genes, we observed KDM4A binding at TSS of AMOT and it moved 500 bp upstream of TSS (TSS-500bp) (Figs 5A and S2A), while it binds at TSS-300bp of MYB and moved to TSS of MYB upon KIKAT/LINC01061 overexpression (S2A and S2B Fig). The reduction of KDM4A occupancy at TSS and increase at TSS-500bp of AMOT promoter upon KIKAT/LINC01061 overexpression was first validated by an independent ChIP experiment (Fig 5B, upper panel). Further analysis of histone marks in control and KIKAT/LINC01061 overexpressing cells by ChIP-qPCR showed that the H3K4me3 was increased at the TSS and TSS-500bp of AMOT after KIKAT/LINC01061 overexpression (Fig 5B, middle panel), while the H3K9me3 levels remained constant at both region (Fig 5B, bottom panel). Consistently, KIKAT/LINC01061 overexpression induced a significant increase of AMOT (Fig 5C and 5D) and knockdown of KIKAT/LINC01061 down-regulated the expression of AMOT (Fig 5E). However, knockdown of KIKAT/LINC01061 has little effect on KDM4A binding (Fig 5F), suggesting KIKAT/LINC0106 does not mainly participate in

**Table 2. KDM4A-KIKAT target in cancer-related pathways.**

| Gene symbol | Distance to TSS (bp) | | Shift Distance (bp) | Cancer pathways |
|---|---|---|---|---|
| | Ctrl | KIKAT | | |
| AMOT | 0 | -558 | -558 | + |
| CCDC184 | 91 | -267 | -358 | + |
| CCDC30 | 147 | -268 | -415 | + |
| CCDC85A | 119 | 552 | 433 | + |
| CCT6B | -164 | 276 | 440 | |
| CITED1 | -1115 | -434 | 681 | + |
| CNKSR2 | 606 | -484 | -1090 | |
| CPNE9 | 249 | -813 | -1062 | |
| DBNDD2 | 592 | -1600 | -2192 | |
| DCLRE1A | -488 | -841 | -353 | + |
| ENOX1 | -48 | 696 | 744 | + |
| FAM189A2 | 322 | -414 | -736 | |
| HIST2H2AC | -573 | 133 | 706 | |
| IL15RA | 687 | 251 | -436 | |
| LARGE1 | 1531 | -461 | -1992 | + |
| LYPD1 | 754 | -295 | -1049 | |
| MYB | -325 | 0 | 325 | + |
| NELL2 | 14 | -1105 | -1119 | + |
| NR5A2 | 1588 | -31 | -1619 | |
| PAOX | -851 | -290 | 561 | |
| PDE5A | 0 | -1533 | -1533 | |
| PTGER4 | -857 | -237 | 620 | |
| PTGES3L-AARSD1 | -299 | -618 | -319 | |
| PTHLH | 1315 | -600 | -1915 | |
| REEP1 | 618 | 182 | -436 | + |
| RMDN2 | 0 | -333 | -333 | |
| ROBO2 | -340 | -1061 | -721 | |
| ROPN1L | 90 | -216 | -306 | |
| SIGIRR | 239 | -1917 | -2156 | |
| SLC17A7 | -1244 | -1692 | -448 | |
| SOCS2 | 677 | -1022 | -1699 | |
| SRGAP3 | 1767 | 968 | -799 | |
| ST3GAL6 | 406 | -418 | -824 | |
| SYT12 | -405 | -1090 | -685 | + |
| TGIF2-RAB5IF | -1243 | 0 | 1243 | |
| TMOD1 | 223 | -91 | -314 | + |
| TRIM46 | 127 | -587 | -714 | |
| TSHZ1 | -132 | 432 | 564 | |
| UVSSA | -825 | -1127 | -302 | |
| ZFP1 | 230 | -365 | -595 | |
| ZNF497 | 566 | -289 | -855 | |
| ZNF517 | 282 | -126 | -408 | |
| ZNF559-ZNF177 | 242 | -202 | -444 | |
| ZNF615 | 46 | -323 | -369 | + |

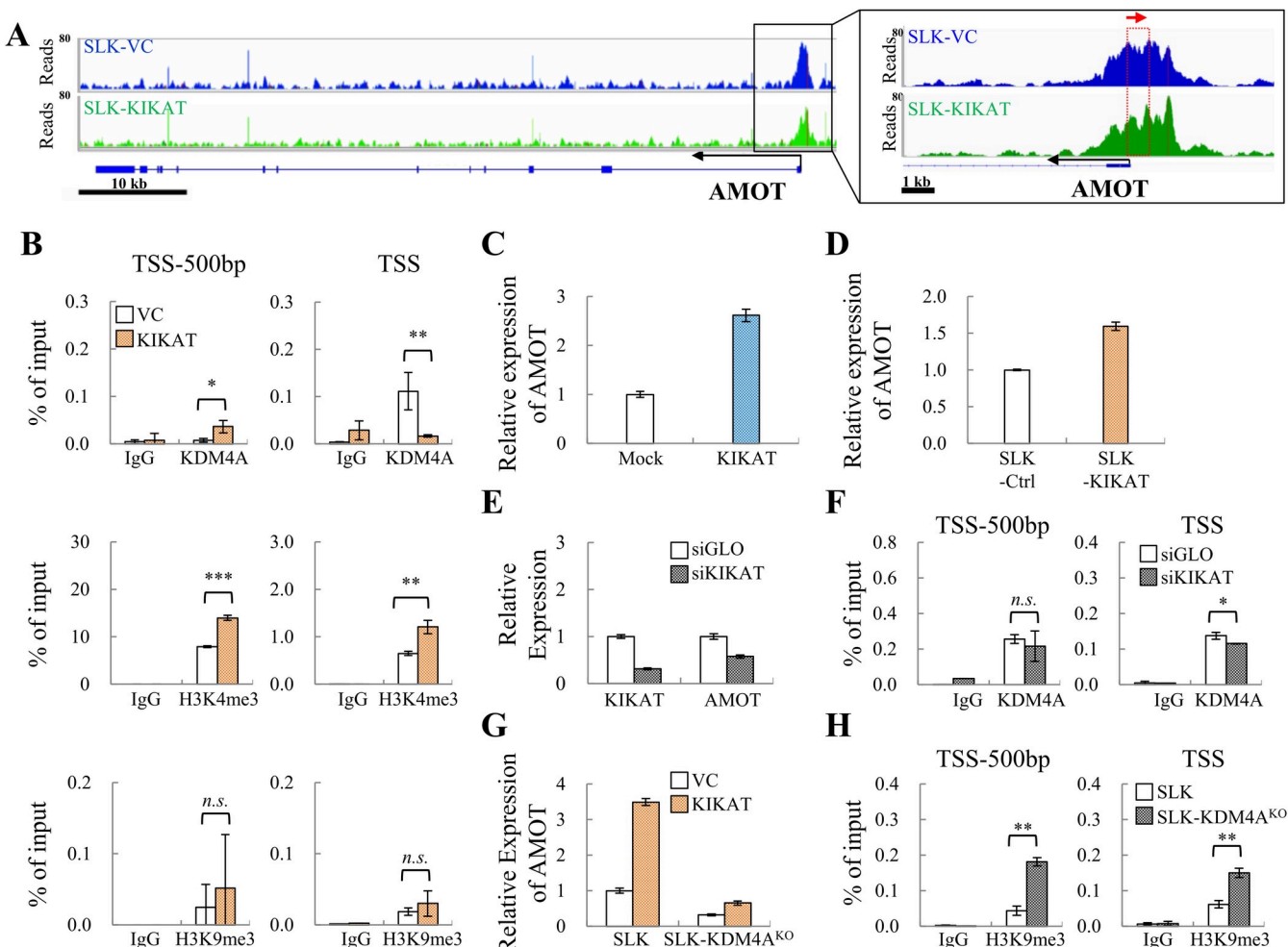

**Fig 5.** Identification of AMOT as a potential KDM4A-KIKAT/LINC01061 target **(A)** Histograms of ChIP-seq profiles for KDM4A binding at AMOT loci in SLK-VC and SLK-KIKAT/LINC01061 cells. Red arrow in the enlarged view shows the direction of KDM4A relocation. **(B)** ChIP-qPCR assay revealed the binding of KDM4A (upper panel) and the modification of H3K4me3 (middle panel) and H3K9me3 (lower panel) to the promoter of AMOT in SLK-VC and SLK-KIKAT/LINC01061 cells. **(C and D)** RT-qPCR analysis verification of AMOT expression in transient KIKAT/LINC01061 transduced SLK cells (C) and in SLK-KIKAT/LINC01061 cell lines (D). **(E)** RT-qPCR analysis revealed the expression of KIKAT/LINC01061 and AMOT in SLK cells after transfected with siRNA targeting KIKAT/LINC01061 and with control siGLO. **(F)** ChIP-qPCR assay revealed the binding of KDM4A to the promoter of AMOT in SLK cells transient transfected with siKIKAT/LINC01061. **(G)** RT-qPCR analysis revealed the expression of AMOT in SLK and SLK-KDM4A KO cells. **(H)** ChIP-qPCR assay revealed the modification of H3K9me3 on the AMOT promoter in SLK-KDM4A KO cells. Significance was determined by student's $t$-test. $^*p < 0.05$, $^{**}p < 0.01$, $^{***}p < 0.001$, *n.s.* non-significant.

KDM4A binding but instead serve as a guide RNA for KDM4A shift on chromatin. The reduction of KDM4A at TSS-300bp and the slight increase at TSS of MYB promoter upon KIKAT/LINC01061 overexpression was also validated by ChIP-qPCR (S2C Fig, upper panel). ChIP-qPCR data showed that the H3K4me3 was reduced at the TSS of MYB promoter (S2C Fig, bottom panel) and the expression of MYB remained un-changed upon KIKAT/LINC01061 over-expression (S2D Fig), indicating that KDM4A move toward TSS may prevent gene from activation. Consistently, KDM4A is required for the up-regulation of AMOT, as KDM4A knockout significantly reduced AMOT up-regulation induced by KIKAT/LINC01061 (Fig 5G). This may due to the increase of H3K9me3 following KDM4A knockout (Fig 5H). Together, these results indicate that KIKAT/LINC01061 facilitates the relocation of KDM4A from TSS region to upstream sequences, and together with KDM4A, protects H3K4me3 from

H3K9me3 invasion, and is required for the transcription of AMOT upon KIKAT/LINC01061 overexpression. However, KIKAT/LINC01061-mediated relocation of KDM4A to TSS at the MYB locus resulted in unchanged gene expression.

## KDM4A-KIKAT/LINC01061 promotes cell migration

Since AMOT encodes angiomotin, an angiostatin-binding protein, which is able to antagonize the inhibitory effect of angiostatin on tube formation and migration of endothelial cells during blood vessel formation [17], we first determined the influence of KIKAT/LINC01061 on cell migration using *in vitro* transwell and wound healing assays. Consistently, KIKAT/LINC01061 overexpression significantly increased cell migration (Fig 6A) and wound healing (Fig 6B) potential of SLK cells. Moreover, knockdown of KIKAT/LINC01061 significantly inhibited cell migration of HMEC1 (Fig 6C) and HUVEC (Fig 6D) cells. We next dissected the contribution of AMOT to KIKAT/LINC01061-induced cell migration. To this end, SLK cells were co-infected with lentiviral vectors expressing KIKAT/LINC01061 and shRNA-targeting AMOT. Transwell assays showed that knockdown of AMOT significantly inhibited SLK migration induced by KIKAT/LINC01061 (Fig 6E and 6F). The essential role of AMOT in KIKAT/LINC01061-induced cell migration was also validated in KSHV containing iSLK-BAC16 cells (Fig 6G and 6H). Together, our data showed that AMOT participates in KIKAT/LINC01061-induced cell migration in both KSHV-free and KSHV containing cells.

## KIKAT/LINC01061 is a K-Rta-induced lncRNA essential for optimal KSHV reactivation

To explore whether KIKAT/LINC01061 is important for KSHV lytic reactivation, the expression kinetics of KIKAT/LINC01061 was first examined in iSLK-BAC16 cells after K-Rta-induced viral reactivation. Immunoblotting and RT-qPCR analysis of KSHV lytic gene products K-bZIP, Orf45, Orf57 and K8.1 after K-Rta induction for 24, 48 and 72 hours revealed the successful induction of viral lytic gene expression (Fig 7A and 7B). Virion associated DNA detection in iSLK-BAC16 cell supernatants using TaqMan qPCR assay [11] indicated the successful induction of KSHV production (Fig 7C). The up-regulation of KIKAT/LINC01061 observed at 24, 48 and 72 hours post-reactivation (Fig 7D) indicated the association of KIKAT/LINC01061 expression and KSHV reactivation. Since K-Rta is a potent transactivator, the up-regulation of KIKAT/LINC01061 upon KSHV reactivation prompted us to speculate that this lncRNA may be a direct target of K-Rta. To study this, the expression level of KIKAT/LINC01061 was examined in iSLK cells after Dox-induced K-Rta overexpression. Consistent with our hypothesis, upon K-Rta overexpression (Fig 7E), KIKAT/LINC01061 was induced (Fig 7F), and transactivation of its promoter by K-Rta was further confirmed by luciferase reporter assays (Fig 7G and 7H).

Next, the impact of KIKAT/LINC01061 on KSHV reactivation was confirmed by knockdown and overexpression experiments. To study this, we transfected siRNA-SMARTpool against KIKAT/LINC01061 into iSLK-BAC16 cells, followed by induction of KSHV reactivation by adding Dox for 48 hours. Remarkably, when expression of KIKAT/LINC01061 was reduced by 30% (Fig 7I, upper panel) in cells during the lytic cycle, viral production was significantly diminished by ~2.5-fold (Fig 7I, lower panel). Furthermore, we transduced iSLK-BAC16 cells with lentivirus expressing KIKAT/LINC01061. Effective overexpression of KIKAT/LINC01061 in iSLK-BAC16 cells was first identified by RT-qPCR (Fig 7J, upper panel). The overexpression of KIKAT/LINC01061 in the lytic cycle only slightly increased the KSHV production (Fig 7J, lower panel), indicating overexpression of KIKAT/LINC01061 alone does not induce KSHV reactivation (Fig 7K). These results together confirm KIKAT/

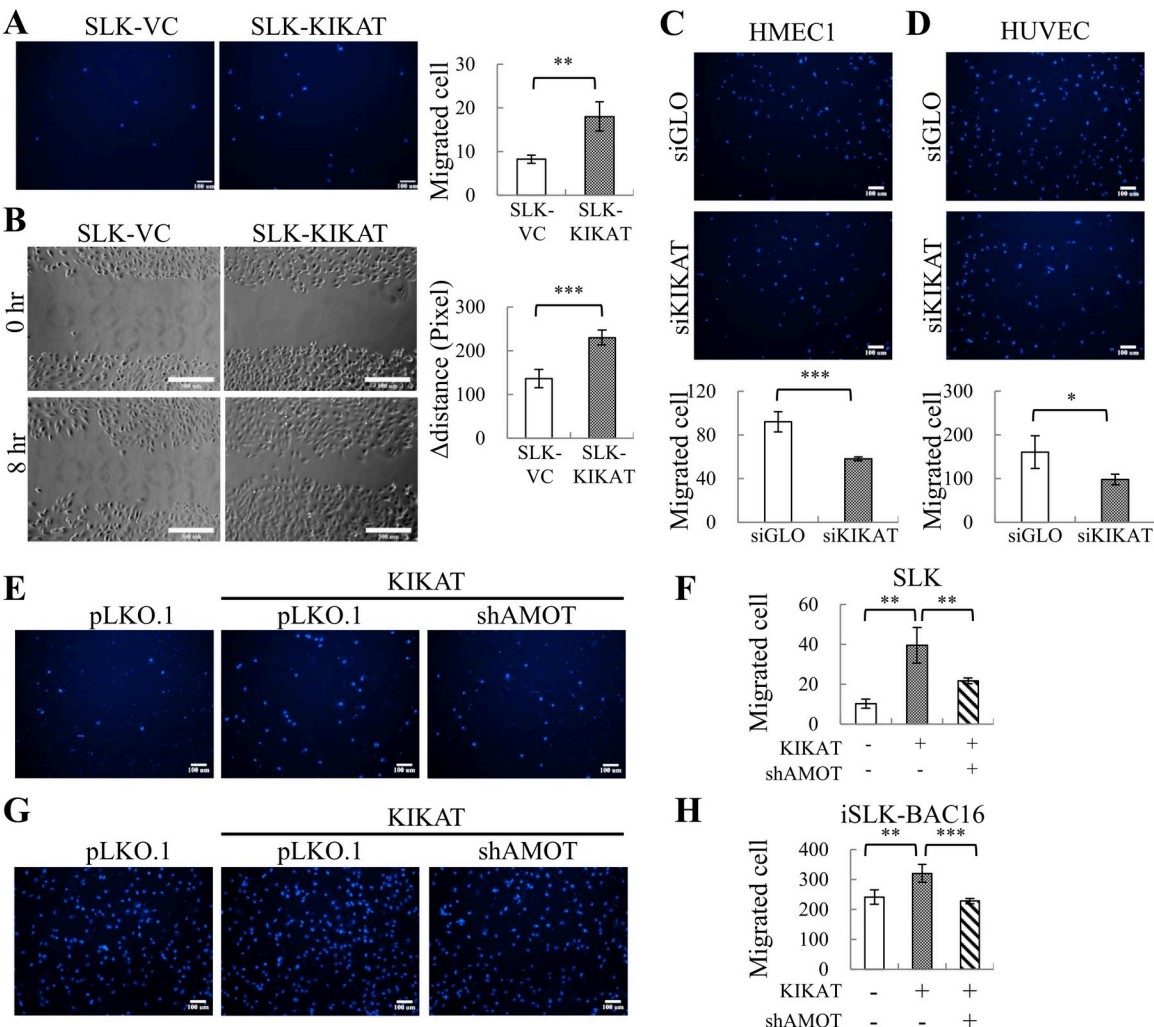

**Fig 6. KIKAT/LINC01061 enhances SLK migration in an AMOT-dependent manner. (A)** Representative images (100x magnification) of *in vitro* transwell migration analysis of SLK-KIKAT/LINC01061 and SLK-VC cells at 6 hours (left). Scale bar: 100 μm. Quantification of cell migration (right). **(B)** Representative images (100x magnification) of *in vitro* wound healing migration assay of SLK-KIKAT/LINC01061 and SLK-VC cells at 8 hours (left panel). Scale bar: 300 μm. Quantification of wound healing (right panel). **(C and D)** Representative images (100x magnification) of migration analysis of HMEC1 (C) and HUVEC (D) cells transfected with siKIKAT/LINC01061 or with control siGLO. (upper panel). Scale bar: 100 μm. Quantification of cell migration (bottom panel). **(E)** Representative images (100x magnification) of migration analysis of KIKAT/LINC01061-transduced SLK cells with or without transduction of lentivirus expressing shAMOT. Scale bar: 100 μm. **(F)** Quantification of cell migration in (E). **(G)** Representative images (100x magnification) of migration analysis of KIKAT/LINC01061-transduced iSLK-BAC16 cells with or without transduction of lentivirus expressing shAMOT. Scale bar: 100 μm. **(H)** Quantification of cell migration in (G). Scale bar: 100 μm. Significance was determined by student's *t*-test. $^{*}p < 0.05$, $^{**}p < 0.01$, $^{***}p < 0.001$.

LINC01061 as a KSHV reactivation-induced lncRNA that is essential for optimal viral reactivation.

## Effect of KIKAT/LINC01061 and KDM4A interaction on KDM4A binding to the KSHV genome

To confirm the epigenetic regulatory role of KIKAT/LINC01061 on KSHV genome, we first conducted RIP assays in control and Dox-treated iSLK-BAC16 cells. Consistently, the interaction between KDM4A and KIKAT/LINC01061 was confirmed (Fig 8A). Next, we assessed

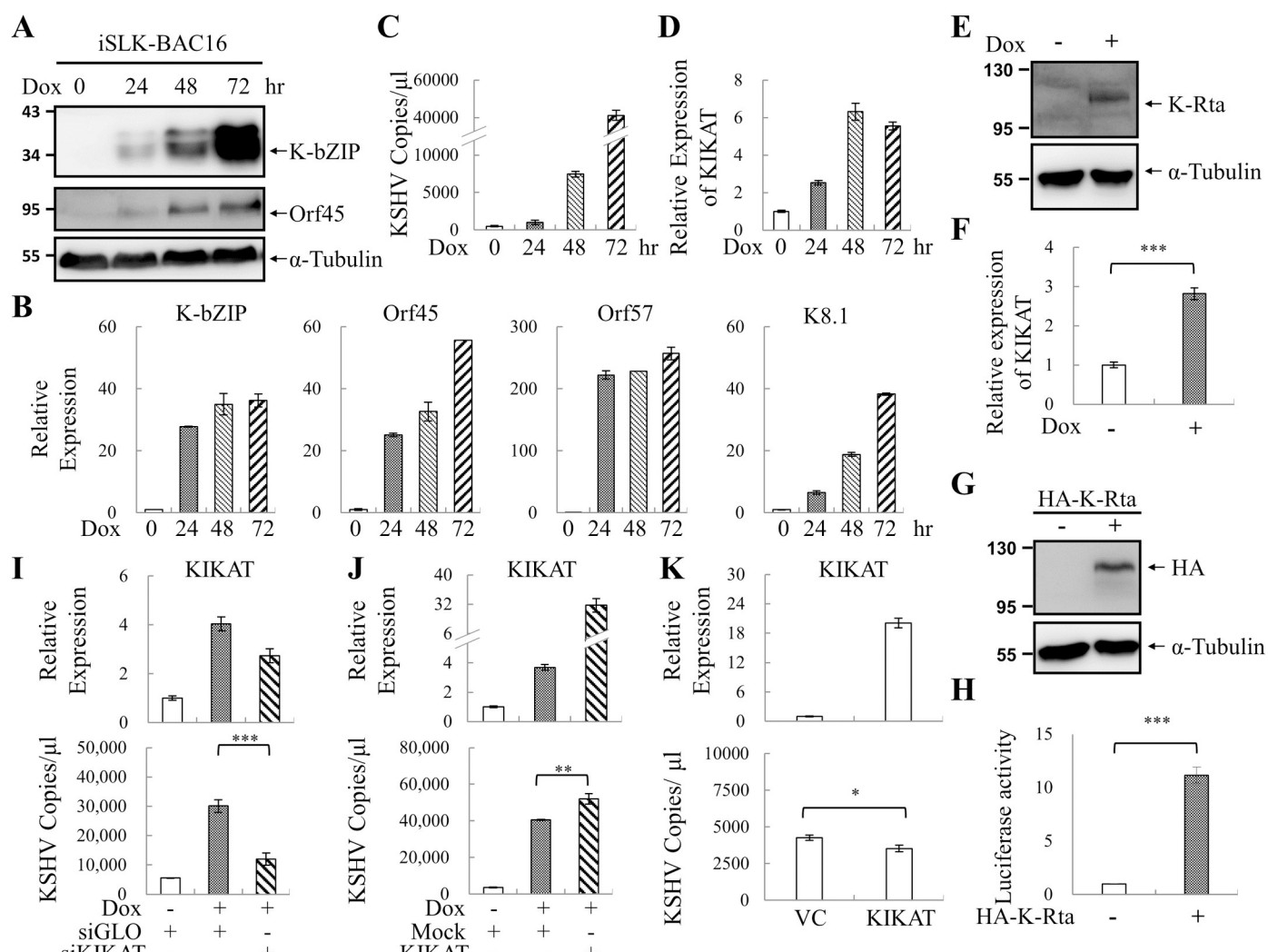

**Fig 7. KIKAT/LINC01061 is important for KSHV reactivation. (A and B)** Immunoblotting (A) and RT-qPCR (B) analysis of the expression of K-bZIP, Orf45, Orf57, K8.1 in iSLK-BAC16 cells treated with 1 μg/ml Dox for 0, 24, 48 and 72 hours. **(C)** KSHV virion-associated DNA was determined by TaqMan qPCR. **(D)** Expression of KIKAT/LINC01061 was determined by RT-qPCR. **(E)** Immunoblotting of K-Rta in iSLK cells treated with or without Dox for 48 hours. **(F)** RT-qPCR analysis of KIKAT/LINC01061 in iSLK cells treated as described in (E). **(G)** The luciferase reporter plasmid containing KIKAT/LINC01061 promoter (TSS ± 500 bp) was co-transfected with pcDNA3-HA-K-Rta or with empty vector into 293T cells. After 48 hours, cells were collected for immunoblotting with anti-HA and anti-α-Tubulin antibodies. **(H)** Luciferase reporter assays were performed in 293T cells treated as described in (G). The activity of firefly luciferase was normalized to that of Renilla luciferase in the same assayed sample and reported as the relative activity of vector control. **(I)** iSLK-BAC16 cells were transfected with siKIKAT/LINC01061 or with control siGLO. After 6 hours, cells were re-seeded in 6-well plates and treated with 1 μg/ml Dox for another 48 hours. Relative expression of KIKAT/LINC01061 (upper panel) and KSHV virion-associated DNA (lower panel) was determined by RT-qPCR and TaqMan qPCR, respectively. **(J)** iSLK-BAC16 cells were transduced with lentivirus expressing full-length KIKAT/LINC01061 or with no cDNA (Mock). After 16 hours, cells were re-seeded in 6-well plates and treated with 1 μg/ml Dox for another 72 hours. Relative expression of KIKAT/LINC01061 (upper panel) and KSHV virion-associated DNA (lower panel) was determined as described in (I). **(K)** iSLK-BAC16 cells were transduced with lentivirus expressing full-length KIKAT/LINC01061 or with no cDNA (Mock). After 72 hours, relative expression of KIKAT/LINC01061 (upper panel) and KSHV virion-associated DNA (lower panel) was determined as described in (I).

chromatin binding alterations of KDM4A in the presence or absence of KIKAT/LINC01061. To achieve this, we performed the ChIP-qPCR in control and Dox-treated iSLK-BAC16 cells with or without siKIKAT/LINC01061 transfection. Consistent with KIKAT/LINC01061 over-expression, KDM4A relocation from TSS to TSS-500bp of AMOT promoter was observed during KSHV reactivation and that was mediated by KIKAT/LINC01061 (Fig 8B). Further analysis of histone marks in control and Dox-treated iSLK-BAC16 cells by ChIP-qPCR showed

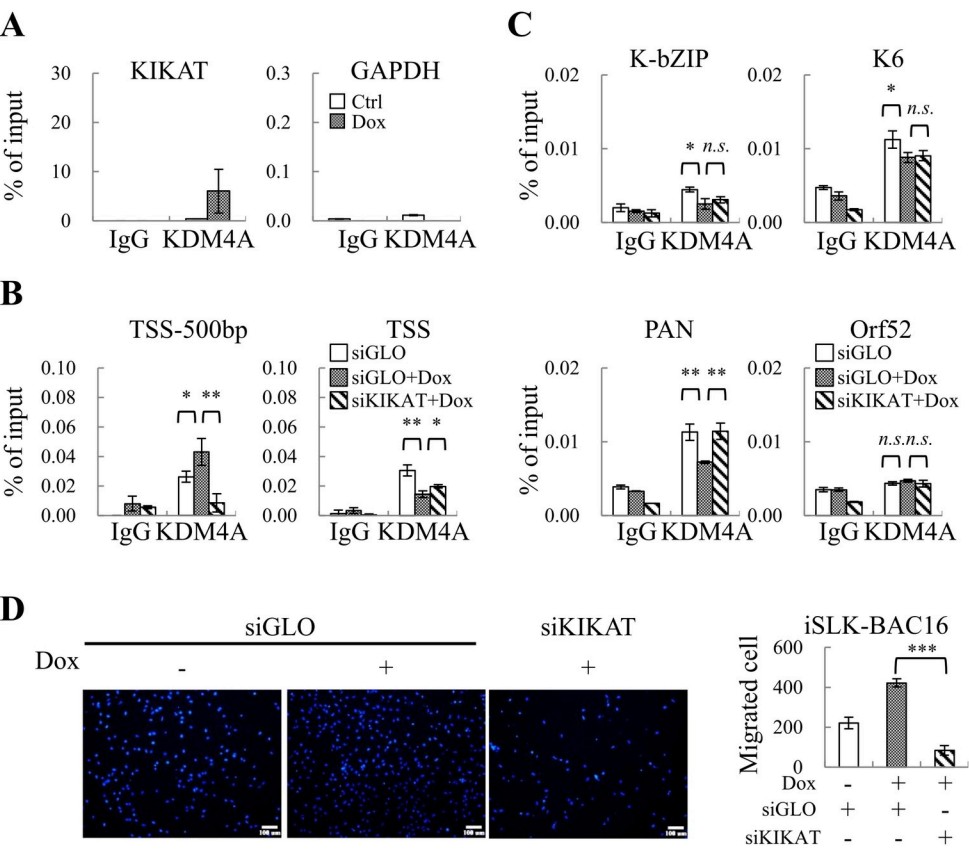

**Fig 8. KIKAT/LINC01061 and KDM4A association in iSLK-BAC16 cells. (A)** TCLs from control and Dox induced iSLK-BAC16 cells were used for RIP using KDM4A antibody or rabbit IgG as control. Copurified RNA from KDM4A and IgG IPs was assessed for KIKAT/LINC01061 enrichment by RT-qPCR. **(B)** Cells as described in Fig 7I were subjected to ChIP assay and the binding of KDM4A to the promoter of AMOT were analyzed by qPCR. **(C)** The enrichment of KDM4A on the promoter of several viral genes in cells described in Fig 7I were analyzed by qPCR. **(D)** iSLK-BAC16 cells were transfected with control siGLO or siKIKAT/LINC01061. After 6 hours, cells were re-seeded in 6-well plate and treated with 1 μg/ml Dox for another 48 hours. *in vitro* transwell migration analysis was determined. Representative images (100x magnification) of transwell migration analysis (left panel). Scale bar: 100 μm. Quantification of cell migration (right panel). Significance was determined by student's *t*-test. *$p < 0.05$, **$p < 0.01$, ***$p < 0.001$.

that the H3K4me3 was only slightly increased at the promoter region of AMOT during KSHV reactivation (S3A Fig, upper panel), while the H3K9me3 levels were significantly reduced in TSS of AMOT (S3A Fig, bottom panel). These results suggest that though KIKAT/LINC01061 may regulate the chromatin binding of KDM4A on AMOT promoter region in a similar fashion with or without KSHV, the histone marks at the examined promoters were differentially regulated upon KSHV reactivation.

We further studied KDM4A binding on KSHV viral promoters that we identified in our previous report [12]. Following reactivation, a reduction of KDM4A binding (Fig 8C) and H3K9me3 mark (S3B Fig, bottom panel) was noted in K-bZIP, K6 and PAN promoter, while the H3K4me3 levels remained constant at these viral gene promoters (S3B Fig, upper panel). More importantly, following reactivation, knockdown of KIKAT/LINC01061 showed little effect on KDM4A binding on the viral promoters examined (Fig 8C). Similar results were obtained in KSHV latently infected BCBL-1 cells (S4 Fig). Together, these data indicate that LINC01061 may have little role in regulating KDM4A binding on the viral genome, but may exerts its effects on KSHV reactivation through actions on host chromatin.

Finally, the role of KIKAT/LINC01061 in transcription regulation and in promoting cell migration was analyzed in the context of KSHV positive cells. For transcription regulation, top 5 up-regulated genes in both transient and stable KIKAT/LINC01061 overexpressed SLK cells (Fig 3C) and in KSHV reactivated iSLK-BAC16 cells (Fig 1A) were analyzed. The up-regulation was demonstrated in four of these genes and knockdown of KIKAT/LINC01061 abolished their induction by KSHV reactivation (S5 Fig), indicating that KIKAT/LINC01061 is essential for transactivation of these genes during viral reactivation. To dissect the contribution of KIKAT/LINC01061 to KSHV reactivation-induced cell migration, we transfected siRNA targeting KIKAT/LINC01061 into iSLK-BAC16 cells, followed by induction of KSHV reactivation by adding Dox for 48 hours. Transwell assays showed that knockdown of KIKAT/LINC01061 significantly inhibited SLK migration induced by viral reactivation (Fig 8D), suggesting KIKAT/LINC01061 also participates in cell migration in KSHV positive cells.

## Discussion

Oncogenic viruses have long served as experimental models for exploring oncogenes and tumor suppressors. KSHV, also known as human herpesvirus type 8 (HHV-8), is one of the seven recognized human cancer viruses [18] and is associated with Kaposi's sarcoma (KS), a highly angiogenic malignancy of endothelial origin [19]. Similar to all herpesviruses, KSHV has two distinct life cycles: latent and lytic, and its life cycle is modulated by epigenetic regulation. Since currently knowledge provide evidence shows that in addition to latent infection, herpesvirus lytic infected cell may also participate in tumorigenesis through an abortive lytic phase that express lytic genes with the potential to cause cancer [20], we have used KSHV reactivation as a model and identified KDM4A as a cancer-associated epigenetic regulator [11,12]. Despite important epigenetic factors having been uncovered by different means in past years, there are still gaps in our knowledge surrounding this form of gene regulation. Recently, emerging evidence has demonstrated that lncRNAs serve as novel epigenetic regulators and play a pivotal role in tumorigenesis. To identify novel lncRNAs associated with chromatin opening, we performed an siRNA screen in rKSHV.219-infected iSLK cells. Successful infection of iSLK cells by rKSHV.219 can be observed by GFP fluorescence and the reactivation of KSHV by K-Rta was identified by RFP fluorescence (Fig 1C). Following siRNA knockdown screening of 82 KSHV reactivation-induced host lncRNAs (Fig 1B), we identified 6 lncRNAs essential for KSHV reactivation (S1 Table and Fig 1D) using GFP:RFP ratio as readout.

KDM4A is the first identified histone trimethyl demethylase with specificity towards the heterochromatin mark trimethyl histone 3 lysine 9 (H3K9me3) [21]. Removal H3K9me3 by KDM4A provides an open chromatin environment, allowing active transcription from promoter sites [22]. The underlying mechanisms of KDM4A in epigenetic regulation have been reported. X-ray crystallography revealed the binding of KDM4A N-terminal catalytic JmjC domain to its substrate H3K9me3 and H3K36me3 [23] and C-terminal tandem Tudor domain (TTD) to chromatin through interacting with H3K4me3 and H4K20me2/3 [24,25]. Given that KDM4A does not possess DNA binding ability, past evidence indicated that target specificity of KDM4A is largely depends on its interacting partners [3,4,26]. In addition, post-translational modifications (PTMs) may also play important role in the chromatin binding of KDM4A. For example, our recent report demonstrated that SUMO modification of KDM4A is required for its chromatin association and gene transactivation [12]. Despite progress that has been made in understanding the chromatin binding and demethylation function of KDM4A, regulation of KDM4A-chromatin interactions still requires further analysis. Interestingly, a recent report showed that KDM4A is important for maintaining proper chromatin boundaries protecting promoter regions with H3K4me3 from H3K9me3 invasion [10].

Consistently, by using KSHV as a model, we have previously showed an inversed chromatin occupancy of KDM4A and H3K9me3 in the KSHV genome [11]. The KDM4A binding regions are also enriched for promoter activation mark H3K4me3 [27], supporting the notion that KDM4A protects H3K9 from being methylated. In the same report, we also demonstrated that KDM4A maintained the viral genome in a "ready to activate" status [11]. However, how KDM4A repositions away from demethylated promoter regions to boundaries and enables transcription activation is largely unknown.

To continue our previous work on KDM4A, here, we focus on studying the KDM4A-interacting lncRNA, KIKAT/LINC01061 (Fig 2). As an oncogenic virus-induced lncRNA (Fig 7), we hypothesized that induction of KIKAT/LINC01061 may be related to tumorigenesis. Consistent with our hypothesis, whole-transcriptome GO analysis revealed significant enrichment of cancer-associated pathways following transient and stable overexpression of KIKAT/LINC01061 in SLK cells (Table 1). In addition, GSEA analysis also demonstrated that KIKAT/LINC01061 expression was correlated with KS lesions (Fig 3H). Consistently, KIKAT/LINC01061 has recently been reported to regulate the oncogenic role of SEMA4D in cholangiocarcinoma [28]. Although, one report showed that DNA copy number gain may mediate KIKAT/LINC01061 up-regulation in papillary thyroid cancer [29], the mechanism contributing to the induction of KIKAT/LINC01061 in different cancers awaits further elucidation. For instance, we found in RT-qPCR analysis that K-Rta overexpression induces the expression of KIKAT/LINC01061 (Fig 7F and 7H), suggesting that perhaps K-Rta may also play a role in host lncRNA activation. These data indicate that KIKAT/LINC01061 maybe up-regulated differently in oncovirus or non-oncovirus-mediated tumorigenesis. Moreover, our previous study showed that KDM4A is SUMO modified by viral SUMO E3 ligase K-bZIP [12]. Post translational modifications (PTMs) has been reported to control the function of an RNA-binding protein [30], including its interaction with RNA [31,32]. However, whether PTMs, such as SUMO modification, regulate the interaction of KDM4A and KIKAT/LINC01061 in oncovirus-mediated tumorigenesis is largely unknown. Therefore, the potential difference of interaction between KIKAT/LINC01061 and KDM4A in oncovirus or non-oncovirus-mediated tumorigenesis is worth further analysis.

As mentioned above, lncRNA has been implicated as a modular scaffold for histone modification complexes. To reveal the epigenetic regulation of KDM4A by KIKAT/LINC01061, we integrated transcriptome profiling and ChIP-seq analyses and identified 145 KIKAT/LINC01061 overexpression up-regulated genes with KDM4A binding in their promoter region (Fig 4C). Interestingly, a shift of KDM4A binding peak (> 300 bp) following KIKAT/LINC01061 overexpression was found in 30% (44/145) of these potential KDM4A targets (Table 2). We therefore suspected that interaction with KIKAT/LINC01061 may account for the movement of KDM4A on the promoter regions. Bioinformatic analysis revealed that 14 of the 44 (32%) potential KDM4A-KIKAT/LINC01061 target genes were found in the cancer-related pathways identified in KIKAT/LINC01061-overexpressing SLK cells (Table 1), and AMOT was one of them (Table 2). Our ChIP-seq data show that KDM4A binds to TSS of the AMOT promoter and this binding shifts 500 bp upstream of the TSS after KIKAT/LINC01061 overexpression (Figs 5A and S2A and Table 2). ChIP-qPCR data showed a reduction of KDM4A and an enrichment of H3K4me3 at the TSS of AMOT, while the H3K9me3 mark remains unchanged (Fig 5B). Consistently, the overexpression and knockdown experiments indicated that KIKAT/LINC01061 positively regulated the expression of AMOT (Fig 5C and 5D).

Angiomotin (AMOT) belongs to the motin family, consisting of 3 members: AMOT (p80 and p130 isoforms), AMOT-like protein 1 (Amotl1), and AMOT-like protein 2 (Amotl2). AMOT was initially identified as an angiostatin-binding protein that promotes endothelial cell migration and angiogenesis [33]. A latter study differentiated the functional role of the two

isoforms of AMOT. The migration-promoting function has been observed in AMOT-p80, but not in AMOT-p130 [34]. Since our RNA-seq data showed that the AMOT-p80 isoform was up-regulated by KIKAT/LINC01061, we further determined the role of KIKAT/LINC01061 on cell migration in the presence or absence of AMOT. Our data showed that KIKAT/LINC01061 overexpression mediated an AMOT-dependent increase in cell migration (Fig 6E–6H).

As mentioned above, KS is a highly angiogenic malignancy [19]. More importantly, AMOT has previously been detected in endothelial cells of vessels within KS, but not in the blood vessel of surrounding tissue [33]. Following our previous study showed that KDM4A is essential for KSHV reactivation, we further investigated if KIKAT/LINC01061 might also regulate chromatin binding of KDM4A upon viral reactivation. To study this, we first performed RIP assays and verified KDM4A associated with KIKAT/LINC01061 after KSHV reactivation (Fig 8A). In addition, we found that the binding of KDM4A was also relocated from TSS to TSS-500bp of AMOT promoter during KSHV reactivation (Fig 8B). However, histone mark alteration at this site was different during viral induction (S3A Fig) when compared to KIKAT/LINC01061 overexpression in uninfected cells (Fig 5B). Together, these data indicated that though KIKAT/LINC01061 may still mediate the KDM4A relocation on chromatin during KSHV reactivation, but different regulators may participate in histone modification in the viral context. KIKAT/LINC01061-KDM4A-mediated epigenetic regulation could be more complex in the presence of KSHV. Moreover, previous reports, in combination with our findings imply that KIKAT/LINC01061-KDM4A mediated induction of AMOT may play a role in angiogenesis of KS, although further experiments are required to confirm this hypothesis.

In this study, we identified KIKAT/LINC01061 as a novel KDM4A-associated lncRNA which is important for KSHV reactivation and cell migration. Mechanistically, KIKAT/LINC01061 is able to interact with KDM4A, facilitates KDM4A relocation from TSS of AMOT promoter and transactivation of AMOT by increasing a histone activation mark and preventing the invasion of a histone repressive mark. We demonstrated that transactivation of AMOT contributed to KIKAT/LINC01061-induced cell motility. Moreover, bioinformatic analysis of our transcriptome data also revealed that KIKAT/LINC01061 is a potential onco-lncRNA associated with KS lesions. Accumulating evidence shows that the epigenetic regulatory protein KDM4A plays an important role in cancer [35] and the potential for targeting KDM4A for cancer therapy exists [36]. In contrast to effects on host chromatin, KIKAT/LINC01061 had little effect on KDM4A binding dynamics on the KSHV genome (Figs 8C and S4), suggesting the KDM4A-KIKAT/LINC01061 interaction augments KSHV reactivation via epigenetic modulation of host cell chromatin. Together, our results demonstrated a novel mechanism for KIKAT/LINC01061-mediated epigenetic regulation of KDM4A and their interaction may serve as a potential target for cancer therapy.

## Materials and methods

### Cell culture

All cells used in this study were cultured at 37°C supplied with 5% $CO_2$. 293T, Vero-rKSHV.219, SLK and SLK-derived cell lines were cultured in Dulbecco's modified Eagle's medium (DMEM, Gibco, 12100–038) containing 10% FBS (Gibco, 26140–079), 1% penicillin-streptomycin (Invitrogen, 1514–0122) and 0.3 mg/ml L-glutamine (Sigma, G8540). The doxycycline (Dox)-inducible iSLK cells were supplemented with 250 μg/ml G418 (Sigma, A1720) and 1 μg/ml puromycin (InvivoGen, ant-pr-1). iSLK cells harboring a KSHV BAC clone, iSLK-BAC16, was maintained as described for iSLK cells and supplemented with 300 μg/ml hygromycin (Gibco, 10687010). Vero-rKSHV.219 cells were supplemented with 5 μg/ml puromycin (InvivoGen, ant-pr-1). Human dermal microvascular endothelial (HMEC1) cells were

maintained in MCDB131 medium (Gibco, 2027–357) supplemented with 10% FBS, 20 ng/ml EGF (PRO-SPEC, cyt-217), 5 μg/ml hydrocortisone and 10 mM L-glutamine. Human umbilical vein endothelial cells (HUVECs) were cultured in EBM-2 Basal Medium (Lonza, CC3156) supplied with EGM-2 SingleQuots Supplements (Lonza, CC4176).

SLK cells stably expressing KIKAT/LINC01061, SLK-KAKIT/LINC01061, were generated by lentiviral transduction. For lentivirus production, plasmids harboring the packaging construct, the envelope-expressing construct and the transfer vector expressing KIKAT/LINC01061 were co-transfected into 293T cells using TransFectin Lipid Reagent (Bio-Rad, 170–3351). After 48 hours, the lentivirus-containing supernatant were collected, filtered by 0.8 μm filter, mixed with 0.8 μg/ml polybrene (Sigma, H9268), and used to infect SLK cells. Forty-eight hours after infection, cells were selected for 1 month by 20 μg/ml blasticidine (InvivoGene, ant-bl-1).

KDM4A knockout cells were generated by targeting the third exon on KDM4A in SLK cells. Plasmids (PX330) expressing guide RNA (gRNA) and Cas9 were transfected into SLK by X-tremeGENE HP (Roche, 6366236001). The sequence of sgRNA targeting exon 3 of KDM4A is as follows: 5'-CCG CAA GAT AGC CAA TAG CGATA-3'. Three days after transfection, cells were subjected to fluorescence-activated cell sorting (FACS) to isolate RFP-positive cells which were diluted and plated in 96-well-plates at 1 cell/well. Immunoblotting assays were used to analyze the expression of KDM4A. Successful genome editing was confirmed by Sanger sequencing of PCR fragment amplified from genomic DNA of selected clones. The primer pair used for PCR amplification was 5'-CTT GCC AGG TTT CTC ATC TGT C-3' and 5'-GGT TTC CAC TCA CTT ATC GCT-3'.

## Expression vectors

KIKAT/LINC01061 cDNA was chemically synthesized and cloned into pUC57 (pUC57-KIKAT/LINC01061) by Genomics (Taipei, Taiwan). KIKAT/LINC01061 were subcloned into lentivector pLenti6-CpoI plasmid modified from pLenti6/TR (Invitrogen, V48020) for KIKAT/LINC01061 overexpression (S6 Fig). AMOT knockdown plasmid (TRCN0000342987) was purchased from RNA Technology and Gene Manipulation Core in Academia Sinica.

## siRNA screening using KSHV reactivation assay

For rKSHV.219 production, Vero-rKSHV.219 were seeded in 10 cm dishes with 60% confluence and treated with 1.75 mM sodium butyrate (NaB) (Sigma, B5887) for 3 days. Supernatants containing rKSHV.219, were collected, filtered and spinoculated onto iSLK cells in the presence of 0.8 μg/ml polybrene. Forty-eight hours after spinoculation, iSLK-rKSHV.219 cells were re-seeded in 96-well plates at a density of 1000 cells/well. Cells were then transfected with 0.5 pmole of siRNA from siLncRNA library (Dharmacon, GU-301000; Lincode Set of Four siRNA Library-Human NR lncRNA RefSeq v54, Lot 13101) or siGLO (Dharmacon, D-001630-01-05) using Lipofectamine RNAiMAX (Invitrogen, 13778–150) according to the manufacturer's protocol. One day after transfection, cells were treated with 2 μg/ml Dox for another 72 hours. Cells were then fixed by 4% paraformaldehyde (Alfa Aesar, 43368), stained with Hoechst 33258 (Invitrogen, H3569), and observed under a fluorescence microscopy (Leica, DMI4000B) with 100x magnification.

## Quantification of KSHV virions by TaqMan qPCR

Supernatants from control and Dox-induced iSLK-BAC16 cells were collected, filtered, and extracted for viral DNA using QIAamp MinElute Virus Spin kits (Qiagen, 57704) following the manufacturer's protocol. Quantification was performed by TaqMan quantitative PCR targeting Orf73 (5'-FAM/TCA GAA CAT CAC CAC CCC ACA GAC/BHQ-3') [37].

## RNA extraction and reverse transcription qPCR (RT-qPCR)

Total RNA was extracted by TRIZol Reagent (Invitrogen, 15596018) and reverse-transcribed by using SuperScript III (Invitrogen, 18080085) for detecting protein-coding mRNA and by using Maxima First Strand cDNA Synthesis Kit (Thermo Scientific, K1642) for detecting non-coding RNA. Quantitative PCR (qPCR) was performed using iTaq Universal SYBR Green Supermix (Bio-Rad, 172–5120) in triplicate, with normalization to GAPDH. Primer sequences are listed in S5 Table.

## RNA-seq analysis

Total RNA was prepared from iSLK-BAC16 cells treated with and without Dox (1 μg/ml), from SLK cells transduced with or without KIKAT/LINC01061, and from SLK-VC and SLK-KIKAT/LINC01061 cells using TRIzol. RNA-seq libraries were prepared using 4 μg of total RNA and the Illumina HiSeq Rapid PE Cluster Kit v2 (Illumina, TG-403-2001) according to the manufacturer's instructions. Sequencing was performed by the sequencing core facility of cancer progression research center at National Yang-Ming University using the HiSeq2000 platform (Illumina). The raw reads were aligned to human genome GRCh37/Hg19 using CLC genomic Workbench v.11 (Qiagen, USA). Transcript levels were expressed as reads per kilobase of transcript per million mapped reads (RPKM) with mRNA and lncRNA information obtained from RefSeq using Partek Genomics Suite (PGS; Partek Inc., St. Louis, Missouri, USA). Differential expression of mRNAs and lncRNAs was analyzed by comparing RPKM and calculated as fold change. The differential expressed mRNAs were subjected to gene ontology (GO) analysis using Ingenuity Pathway Analysis (IPA) software (QIAGEN, Hilden, Germany).

Human Cancer Metastasis DataBase (HCMDB) was applied to assess the correlation of KIKAT/LINC01061 expression level with tumor progression (http://hcmdb.i-sanger.com/index). GSEA (Gene Set Enrichment Analysis) was applied to identify the gene set enriched in the sub-clusters of 26 skin cancer samples (GSE7553) and 24 KS lesion samples (GSE147704). The 245 genes that are repressed by at least 5-fold in SLK-KIKAT/LINC01061 cells when compared with control were used for GSEA analysis. Typically, significant thresholds were defined as $q$ (false discovery rate) $< 0.25$ and $p < 0.05$.

## Chromatin immunoprecipitation-sequencing (ChIP-seq) and real-time qPCR

ChIP was performed according to the protocol from Dr. Farnham's laboratory (http://genomics.ucdavis.edu/farnham). Chromatin DNA prepared from $1 \times 10^7$ of SLK-VC, SLK-KIKAT/LINC01061, and SLK-KDM4A KO cells was used for each ChIP experiments. 1 μg of anti-KDM4A, anti-H3K4me3 (Millipore, 07–473), anti-H3K9me3 (Abcam, ab8898) rabbit polyclonal antibodies and anti-rabbit IgG were used for the ChIP assays. ChIP-seq libraries were prepared using 10 ng of purified DNA and the Illumina HiSeq Rapid PE Cluster Kit v2 (Illumina, TG-403-2001) according to the manufacturer's instruction. Sequencing was performed as described in RNA-seq. The raw reads were trimmed and aligned to human genome GRCh37/Hg19. ChIP-seq peaks were called by PGS using mock infected SLK-KDM4A KO cells as reference.

## RNA immunoprecipitation (RIP)

RIP was performed using Magna RIP RNA-Binding Protein Immunoprecipitation kit (Millipore, 17–700) according to the manufacturer's instructions. $1 \times 10^7$ cells were used for each

RIP experiment. The RNA pellets were dissolved in 10 μl nuclease-free water and 5 μl of RNA were reverse transcribed by Maxima First Strand cDNA Synthesis Kit.

## Immunoblotting analysis

Cells were collected in PBS and lysed in PBS containing 0.5% Nonidet P-40 (NP-40) containing 1X protease inhibitor cocktail (Roche, 4693132001). The cell lysates were sonicated 5 times (pulsed for 30 sec with 30 sec interval) using a Bioruptor (Diagenode, UCD-200) at low power setting. Total cell lysate (TCL) were separated by sodium dodecyl sulfate polyacrylamide gel electrophoresis (SDS-PAGE), transferred to polyvinylidene fluoride membranes (PVDF; PerkinElmer, 1002001), blocked with 5% nonfat skim milk or 5% BSA (UniRegion, UR-BSA001-50G) in 1X TBST, and probed with primary antibody at 4 °C for 16 hours. After incubating the membrane with horseradish peroxidase (HRP) linked secondary antibody (GE healthcare, NA934 and NA931), the protein expression levels were detected by Pierce ECL Western Blotting Substrate (Thermo, 32106). Primary antibodies against KDM4A (Polyclonal antibody purified from rabbit) [12], K-bZIP (1:1000; Santa Cruz Biotechnology, sc-69797), Orf45 (1:1000; Santa Cruz Biotechnology, sc-53883), HA tag (1:4000; Cell Signaling Technology, #3724) and α-Tubulin (1:4000; Sigma, T6074-200UL) were used in this study.

## Transwell assay

SLK and iSLK-BAC16 cells were seeded on upper chamber ($2 \times 10^4$ and $5 \times 10^4$ cells/well, respectively) of 24-well transwell plates (8 μm) (Costar, 3422) in 200 μl of serum-free DMEM containing 20 ng/ml epidermal growth factor (EGF; PROSPEC, cyt-217), 10 ng/ml fibroblast growth factor basic (FGF-b; PROSPEC, cyt-218) and B27 (Gibco, 17504044). HMEC1 ($1 \times 10^4$ cells/well) and HUVEC ($1 \times 10^4$ cells/well) were seeded in serum free MCDB131 and 2% serum EGM-2. Cells were allowed to migrate to the lower chamber filled with 1 ml cultured medium containing 10% FBS for 6 hours at 37˚C. After incubation, cells from unmigrated (top) side of transwell were gently removed by a cotton swab. Migrated cells were fixed by 4% paraformaldehyde for 15 minutes at 37˚C and stained with Hoechst 33258. The migrated cells were quantified by photographing 4 independent fields under the fluorescence microscopy using MetaMorph Microscopy Automation and Image Analysis Software (Molecular Devices, Downingtown, PA).

## Wound healing assay

SLK cells were seeded into Ibidi Culture-Inserts (Ibidi, 80209) at a density of $3 \times 10^4$ cells/well. After 16–18 hours, Culture-Inserts were removed and cells were cultured in 1 ml DMEM containing 10% FBS for 8 hours at 37˚C. The area of the wound was observed under a microscope.

## Supporting information

**S1 Fig.** Generation of SLK KDM4A knockout cell line by CRISPR/Cas9 system **(A)** Schematic illustrating DNA breaks using Cas9 D10A nickases (Cas9n). The sequence of guide RNA targeting exon 3 of KDM4A (Cas9 KDM4A E3g2) was showed as follows: 5'-TAT CGC TAT TGG CTA TCT TGC GG-3'. **(B)** Immunoblotting of KDM4A in different KDM4A KO clones. **(C)** The knockout clones (g2-2, g2-3, g2-5, and g2-6) were confirmed by sequencing. Deleted sequences are indicted by dotted lines (Upper panel). The protospacer-adjacent motif (PAM) sequence is labeled in red rectangle. Chromatograms from Sanger sequencing of PCR amplicons over exon 3 of KDM4A are shown (Lower panel).
(TIF)

**S2 Fig. Identification of KDM4A binding on KDM4A-KIKAT/LINC01061 targets following KIKAT/LINC01061 overexpression.** **(A)** Heatmap of ChIP-seq read density of KDM4A peaks in promoter region (TSS ± 2,000 bp) of KIKAT/LINC01061-regulated KDM4A-targeted genes. **(B)** Histograms of ChIP-seq profiles for KDM4A binding at MYB loci in SLK-VC and SLK-KIKAT/LINC01061 cells. **(C)** ChIP-qPCR assay revealed the binding of KDM4A (upper panel) and the modification of H3K4me3 (lower panel) to the promoter of MYB in SLK-VC and SLK-KIKAT/LINC01061 cells. **(D)** RT-qPCR analysis of MYB expression in SLK-KIKAT/LINC01061 cell lines.
(TIF)

**S3 Fig.** Histone marks alteration during KSHV reactivation (A) ChIP-qPCR assay revealed the modification of H3K4me3 (upper panel) and H3K9me3 (bottom panel) to the promoter of AMOT in iSLK-BAC16 cells treated with or without Dox for 72 hours. (B) ChIP-qPCR assay revealed the enrichment of H3K4me3 (upper panel) and H3K9me3 (lower panel) to the promoter of viral genes in iSLK-BAC16 cells before and after Dox induced KSHV reactivation for 72 hours.
(TIF)

**S4 Fig. Identification of KDM4A binding on latent KSHV genome following knockdown of KIKAT/LINC01061.** ChIP-qPCR assay revealed no changes in the enrichment of KDM4A on the promoter of select viral genes in TREx-F3H3-K-Rta BCBL-1 cells during latency before and after knockdown of KIKAT/LINC01061.
(TIF)

**S5 Fig. Identification and verification of KIKAT/LINC01061-activated genes in iSLK-BAC16.** **(A)** Expression heatmap of top 5 up-regulated genes of RNA-seq data from KSHV reactivated iSLK-BAC16 cells and from transient and stable KAKIT/LINC01061 overexpression SLK cells. **(B)** RT-qPCR analysis of the expression of top 5 up-regulated genes identified in (A) in iSLK-BAC16 cells. **(C)** iSLK-BAC16 cells were transfected with siKIKAT/LINC01061 or with control siGLO. After 6 hours, cells were re-seeded in 6-well plate and treated with 1 µg/ml Dox for another 48 hours. RT-qPCR analysis of the 4 up-regulated genes identified in (B).
(TIF)

**S6 Fig. Construction of pLenti6-KIKAT/LINC01061 plasmid.** The pLenti6/TR lentiviral vector (Invitrogen, V48020) purchased from ThermoFisher was digested with *Xba*I to remove the TR, followed by insert a linker containing restriction site for *Cpo*I. The KIKAT/LINC01061 cDNA digested by *Cpo*I was cloned into the pLenti6-*Cpo*I plasmid digested with the same enzyme.
(TIF)

**S1 Table. siRNA screening of KSHV reactivation-induced lncRNAs essential for rKSHV.219 latent-to-lytic switch.**
(XLSX)

**S2 Table. Differential expressed genes in transient KIKAT/LINC01061 overexpression cells.**
(XLSX)

**S3 Table. Differential expressed genes in KIKAT/LINC01061 stable overexpression cell line.**
(XLSX)

**S4 Table. ChIP-Seq and RNA-Seq analyses identify KIKAT/LINC01061-regulated KDM4A-targeted genes.**
(XLSX)

**S5 Table. Primer sequences used for qPCR.**
(XLSX)

**S6 Table. The numerical data used in all figures.**
(XLSX)

## Acknowledgments

The authors would like to thank Dr. Hsing-Jien Kung (University of California, Davis, USA) for Lincode Set of Four siRNA Library (Dharmacon, GU-301000), Dr. Szu-Ting Chen (National Yang-Ming University, Taipei, Taiwan) for HMEC1 cell line, and Dr. Tze-Tze Liu (National Yang-Ming University, Taipei, Taiwan) for high-throughput sequencing data generation. The authors acknowledge the High-throughput Genome and Big Data Analysis Core Facility of National Core Facility Program for Biotechnology, Taiwan (MOST 105-2319-B-010-001) for sequencing.

## Author Contributions

**Conceptualization:** Wan-Shan Yang.

**Data curation:** Wan-Shan Yang, Wayne W. Yeh.

**Formal analysis:** Wan-Shan Yang, Wayne W. Yeh, Pei-Ching Chang.

**Funding acquisition:** Pei-Ching Chang.

**Investigation:** Wan-Shan Yang, Wayne W. Yeh, Lung Chang.

**Methodology:** Wan-Shan Yang, Wayne W. Yeh, Mel Campbell, Lung Chang.

**Project administration:** Wan-Shan Yang, Pei-Ching Chang.

**Resources:** Lung Chang, Pei-Ching Chang.

**Software:** Wan-Shan Yang, Wayne W. Yeh, Pei-Ching Chang.

**Supervision:** Wan-Shan Yang, Pei-Ching Chang.

**Validation:** Wayne W. Yeh, Mel Campbell, Lung Chang.

**Writing – original draft:** Wan-Shan Yang, Pei-Ching Chang.

**Writing – review & editing:** Mel Campbell, Pei-Ching Chang.

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
