## [Decision Letter · Decision Letter 0]

10 Feb 2021

Dear phD Chang,

Thank you very much for submitting your manuscript "A herpesvirus screen to identify long non-coding RNA important for viral reactivation connects LINC01061 as an epigenetic regulator that relocates KDM4A on chromatin" for consideration at PLOS Pathogens. As with all papers reviewed by the journal, your manuscript was reviewed by members of the editorial board and by several independent reviewers. In light of the reviews (below this email), we would like to invite the resubmission of a significantly-revised version that takes into account the reviewers' comments.

We cannot make any decision about publication until we have seen the revised manuscript and your response to the reviewers' comments. Your revised manuscript is also likely to be sent to reviewers for further evaluation.

Sincerely,

Fanxiu Zhu, Ph.D.

Associate Editor

PLOS Pathogens

Erik Flemington

Section Editor

PLOS Pathogens

Kasturi Haldar

Editor-in-Chief

PLOS Pathogens

orcid.org/0000-0001-5065-158X

Michael Malim

Editor-in-Chief

PLOS Pathogens

orcid.org/0000-0002-7699-2064

Reviewer's Responses to Questions

**Part I - Summary**

Reviewer #1: This is a resubmission of the manuscript on a KSHV RTA-induced noncoding RNA that is important for viral reactivation and acts as an epigenetic regulator by relocating KDM4A on chromatin. Authors have addressed most of the questions raised by reviewers. The revised manuscript is improved. The followings are some suggestions that may help to further strengthen the manuscript.

Reviewer #2: The authors addressed several of the questions raised by reviewer 3 but the main concern remained that the results obtained in the KSHV-free system are not connected enough to KSHV biology. The goal of the study was to identify lncRNAs that can control KSHV latent-lytic switch. Several lncRNAs were identified using genomics, which was great. But after that we did not learn anything about how LINC01061 promotes KSHV lytic reactivation, which diminishes the significance of their study, which otherwise generated a number of interesting and exciting data in KSHV-free cells.

The study has basically three findings:

1. LINC01061 is needed to facilitate KSHV lytic reactivation.

2. KDM4A interacts with LINC01061 and LINC01061 overexpression shifts KDM4A-binding on cellular chromatin and affects H3K4m3/H3K9me3 ratio.

3. KDM4A/LINC01061 is needed for AMOT expression, which enhances cell migration.

Reviewer #3: In this manuscript, Yang et al described that a long non-coding RNA (lncRNA) LINC01061 is induced by KSHV in lytic replication, and LINC01061 interacts with KDM4A by RNA immunoprecipitation assay, and transactivates AMOT expression. Knockdown of AMOT reduces the migration ability of SLK cells with LINC01061 overexpression. Therefore, the authors concluded that dysregulation of LINC01061 expression may be a novel pathological mechanism for KDM4A oncogenecity. Overall, the manuscript is improved a lot and presents an interesting discovery in compared with the early version, however, there is still absence of some evidence to support the conclusion.

**Part II – Major Issues: Key Experiments Required for Acceptance**

Reviewer #1: (1) The title indicates that the non-coding RNA is important for viral reactivation. However, the whole article does not address if LINC01061 has any role in KSHV reactivation. Authors have done RNA-seq on LINC01061-over-expressed KSHV-SLK cells and ChIP-seq for KDM4A. Why don’t authors map the reads to the KSHV genome? The results will show the effects of the noncoding RNA on viral gene expression and if LINC01061 relocates KDM4A on the viral chromaton.

(2) Fig. 6A and Fig. S3B, the KDM4A relocation on AMOT and MYB genes needs to be indicated by arrows.

(3) The manuscript can be written more concisely, which will improve the readability of the manuscript. The title needs to be revised.

Reviewer #2: Based on their current findings it is unclear how much of the results from LINC01061 overexpression/knockdown experiments in KSHV-free cells are applicable to KSHV-infected cells. The question remains how many of the genes, which are regulated by LINC01061, are also controlled by LINC01061 in KSHV-infected cells? AMOT is regulated by KDM4A/LINC01061 in SLK cells and affects angiogenesis but not in KSHV-infected cells? RNA-seq analysis of lytic reactivated KSHV-infected cells in which LINC01061 expression is depleted by shRNA could reveal how LINC01061 contributes to KSHV life cycle and the expression of tumorigenesis pathway genes in KSHV+ cells.

Also, the group has published excellent studies on how KDM4A binds to the KSHV genome and affects lytic gene expression. Does LINC01061 also affect KDM4A and H3K4me3/H3K9me3 enrichment/ratio on the viral chromatin like on the host genome?

Reviewer #3: 1) The title of manuscript is not appropriately to summarize and highlight what the current data has presented. Since the majority study was started by lncRNA screening of KSHV lytic reactivation, and then found that the lncRNA LINC01061 was induced by KSHV lytic reactivation for binding to KDM4A and relocation of KDM4A for increasing AMOT expression and migration in SLK cells without KSHV infection. Due to KSHV lytic replication will lead to host cell destroy, why KSHV needs to induce LINC01061 expression in lytic replication instead of latency for oncogenesis? To highlight the physiological relevance of viral pathogenesis or just KDM4A oncogenicity, it would better to focus the role in viral reactivation or host cell effect only, but not both.

2) In Fig.2B and D, it needs to compare the difference of KDM4A and its ability bound to LINC01061 before and after KSHV lytic reactivation, to confirm the conclusion that LINC01061 is specifically induced by KSHV reactivation and bound to KDM4A.

3) In Fig.6, it needs to include the data to prove that KDM4A is also relocated on host cell chromatin after KSHV reactivation, not just SLK-VC and SLK-LINC01061 cell lines.

4) In Fig. 7B, the author presented that LINCO1061 plays a role in cell migration by using SLK cell line. What is the role for KSHV to enhance LINC01061 expression during lytic replication? also enhancing lytic host cell migration? need to be addressed.

**Part III – Minor Issues: Editorial and Data Presentation Modifications**

Reviewer #1: (No Response)

Reviewer #2: Fig 1E, F Some comment would be useful on why most of the lncRNAs that are induced in RTA-expressing iSLK-BAC16 and BCBL1 were not upregulated in HMEC1-KSHV.r219 cells reactivated by RTA. Add these comments after these result were described.

P6 line 36 KHSV-DNA, misspelling of KSHV

Fig 3C ORF57 western blot did not seem to work. I suggest that it be repeated or it can also be deleted.

P6-7 “To this end, we first detected the KSHV titer in iSLK-BAC16 cell supernatants after K-Rta induction for 24, 48 and 72 hours (Fig. 3E).” This requires some conclusion before talking about the follow-up experiment.

P7 line 6-7 “…we further generated an inducible LINC01061 expressing cell line using iSLK-BAC16 cells, iSLK-BAC16-LINC01061.” As far as I understood the lncRNA was overexpressed in iSLK-BAC16 using lentiviral transduction and lytic reactivation was triggered by Dox so the lncRNA was not inducible. If so, I suggest that the sentence be changed.

Fig 4D-E The gene PALM seems to be strongly upregulated by LINC01061 based on the heat map but its RT-qPCR test shows no PALM expression upon LINC01061.

P7 line 28 I believe Fig 4E wants to show the confirmation of host genes that were upregulated by LINC01061. If so, “…was first checked by…” it should read “…was confirmed by…”

P8 “…was also used to assess the tumorigenesis of LINC01061-repressed genes” it should read “…was also used to assess the association of LINC01061-repressed genes with tumorigenesis”

Reviewer #3: (No Response)

PLOS authors have the option to publish the peer review history of their article (what does this mean?). If published, this will include your full peer review and any attached files.

Reviewer #1: No

Reviewer #2: No

Reviewer #3: No
---

## [Decision Letter · Decision Letter 1]

3 May 2021

Dear phD Chang,

Thank you very much for submitting your manuscript "Long non-coding RNA KIKAT/LINC01061 as a novel epigenetic regulator that relocates KDM4A on chromatin and modulates viral reactivation" for consideration at PLOS Pathogens. As with all papers reviewed by the journal, your manuscript was reviewed by members of the editorial board and by several independent reviewers. The reviewers appreciated the attention to an important topic. Based on the reviews, we are likely to accept this manuscript for publication, providing that you modify the manuscript according to the review recommendations.

Sincerely,

Fanxiu Zhu, Ph.D.

Associate Editor

PLOS Pathogens

Erik Flemington

Section Editor

PLOS Pathogens

Kasturi Haldar

Editor-in-Chief

PLOS Pathogens

orcid.org/0000-0001-5065-158X

Michael Malim

Editor-in-Chief

PLOS Pathogens

orcid.org/0000-0002-7699-2064

Reviewer Comments (if any, and for reference):

Reviewer's Responses to Questions

**Part I - Summary**

Reviewer #1: In this revision, the authors have addressed most of the concerns raised in the last submission. The data support the conclusion that LncRNA KIKAT is an epigenetic regulator that modulates viral reactivation and oncogenic phenotypes through binding to KDM4A. The manuscript sounds well in science, but the manuscript can be further polished to make it more concise and improve its readability. Figures 7 and 8 can be combined as there are too little data in each figure.

Reviewer #2: The manuscript is improved after revision and the authors have performed additional experiments to address previous reviewers' question. One of the major questions was how LINC01061 affects KDM4A on the KSHV genome. Figure 9 and Figure S3 provided some new information but it is confusing. It seems that the siRNA kd of LINC01061 increases KDM4A binding at PAN promoter while the binding of KDM4A does not change at other viral promoters upon lytic reactivation. However, H3K9me3 is reduced at PAN while KDM4A binding increases. This is counterintuitive. In JVI 2011 Chang et al paper on JMJD2A/KDM4A, the binding of KDM4A was studied only during latency and in BCBL1. In the current manuscript the authors show in Figure 2E that KDM4A interacts with LINC01061 in BCBL1 during latency. Can they use latent BCBL1 cells and siRNA for LINC01061 to show if LINC01061 affects KDM4A binding to viral promoters that were shown in the JVI 2011 paper Chang et al? If LINC01061 does not affect KDM4A binding on the viral genome, it would indicate that LINC01061 may regulate KDM4A on host but not on viral genome, which is okay. This needs to be addressed in the discussion, which was omitted in the revised manuscript.

Reviewer #3: The authors have addressed most of concerns raised by the reviewers in the third revised manuscript, and it is now suitable for publication in the current version.

**Part II – Major Issues: Key Experiments Required for Acceptance**

Reviewer #1: (No Response)

Reviewer #2: (No Response)

Reviewer #3: (No Response)

**Part III – Minor Issues: Editorial and Data Presentation Modifications**

Reviewer #1: (No Response)

Reviewer #2: (No Response)

Reviewer #3: (No Response)

PLOS authors have the option to publish the peer review history of their article (what does this mean?). If published, this will include your full peer review and any attached files.

Reviewer #1: No

Reviewer #2: No

Reviewer #3: No

Figure Files:

Data Requirements:

Reproducibility:

References:

---

## [Editor Report · Decision Letter 2]

26 May 2021

Dear phD Chang,

We are pleased to inform you that your manuscript 'Long non-coding RNA KIKAT/LINC01061 as a novel epigenetic regulator that relocates KDM4A on chromatin and modulates viral reactivation' has been provisionally accepted for publication in PLOS Pathogens.

Best regards,

Fanxiu Zhu, Ph.D.

Associate Editor

PLOS Pathogens

Erik Flemington

Section Editor

PLOS Pathogens

Kasturi Haldar

Editor-in-Chief

PLOS Pathogens

orcid.org/0000-0001-5065-158X

Michael Malim

Editor-in-Chief

PLOS Pathogens

orcid.org/0000-0002-7699-2064
---

## [Editor Report · Acceptance letter]

8 Jun 2021

Dear phD Chang,

We are delighted to inform you that your manuscript, "Long non-coding RNA KIKAT/LINC01061 as a novel epigenetic regulator that relocates KDM4A on chromatin and modulates viral reactivation," has been formally accepted for publication in PLOS Pathogens.

Best regards,

Kasturi Haldar

Editor-in-Chief

PLOS Pathogens

orcid.org/0000-0001-5065-158X

Michael Malim

Editor-in-Chief

PLOS Pathogens

orcid.org/0000-0002-7699-2064